# 120 GOPS Photonic tensor core in thin-film lithium niobate for inference and in situ training

Zhongjin Lin[1,2], Bhavin J. Shastri [3], Shangxuan Yu[1], Jingxiang Song[1], Yuntao Zhu[2], Arman Safarnejadian[4], Wangning Cai[1], Yanmei Lin[2], Wei Ke[2], Mustafa Hammood[1], Tianye Wang[1], Mengyue Xu [2], Zibo Zheng[4], Mohammed Al-Qadasi[1], Omid Esmaeeli[1], Mohamed Rahim[5], Grzegorz Pakulski[5], Jens Schmid [5], Pedro Barrios[5], Weihong Jiang[5], Hugh Morison [3], Matthew Mitchell [1], Xun Guan[6], Nicolas A. F. Jaeger[1], Leslie A. Rusch [4], Sudip Shekhar [1], Wei Shi [4], Siyuan Yu[2], Xinlun Cai [2] ✉ & Lukas Chrostowski[1] ✉

Photonics offers a transformative approach to artificial intelligence (AI) and neuromorphic computing by enabling low-latency, high-speed, and energy-efficient computations. However, conventional photonic tensor cores face significant challenges in constructing large-scale photonic neuromorphic networks. Here, we propose a fully integrated photonic tensor core, consisting of only two thin-film lithium niobate (TFLN) modulators, a III-V laser, and a charge-integration photoreceiver. Despite its simple architecture, it is capable of implementing an entire layer of a neural network with a computational speed of 120 GOPS, while also allowing flexible adjustment of the number of inputs (fan-in) and outputs (fan-out). Our tensor core supports rapid in-situ training with a weight update speed of 60 GHz. Furthermore, it successfully classifies (supervised learning) and clusters (unsupervised learning) 112 × 112-pixel images through in-situ training. To enable in-situ training for clustering AI tasks, we offer a solution for performing multiplications between two negative numbers.

Artificial intelligence (AI) is increasingly being integrated into various sectors, including autonomous vehicles, smart buildings, and smart factories, as illustrated in Fig. 1a. At the heart of AI systems are tensor core processors, which are expected to exhibit several key characteristics:

(1) High-speed, large-scale matrix-vector multiplication: These processors must efficiently process data from a variety of devices for tasks such as classification (supervised learning) and clustering (unsupervised learning), as depicted in Fig. 1a. Crucially, they perform matrix-vector multiplication and dynamically adjust the input (fan-in) and output (fan-out) sizes of each network layer as needed.

(2) Rapid weight updates: Accurate and efficient training necessitates the use of in situ training[1–4]. This method incorporates real-time feedback from processors into the weight update loop,

[1]Department of Electrical and Computer Engineering, The University of British Columbia, Vancouver, British Columbia, Canada. [2]State Key Laboratory of Optoelectronic Materials and Technologies, School of Electronics and Information Technology, Sun Yat-sen University, Guangzhou, Guangdong, China. [3]Department of Physics, Engineering Physics and Astronomy, Queen's University, Kingston, Ontario, Canada. [4]Department of Electrical and Computer Engineering, Université Laval, Québec City, Québec, Canada. [5]Advanced Electronics and Photonics Research Centre, National Research Council, Ottawa, Ontario, Canada. [6]Tsinghua Shenzhen International Graduate School, Tsinghua University, Shenzhen, China. ✉e-mail: caixlun5@mail.sysu.edu.cn; lukasc@ece.ubc.ca

**Fig. 1 | Concept of our integrated photonic tensor core (IPTC). a** Top: Applications and functions of artificial intelligence (AI)[44,54–56]. AI systems require processors to be adaptable to analyze data from different devices for various AI tasks, including supervised and unsupervised learning AI tasks. Bottom: A schematic of our proposed IPTC, consisting of four physical components: lasers, two thin-film lithium niobate (TFLN) Mach-Zehnder modulators (MZMs), and charge-integration photoreceivers. Using these four physical components, our processor can implement an entire layer of a neural network. **b** A schematic of a conventional wavelength-division multiplexing (WDM)-based IPTC, which includes $m$ neurons. PCM: phase change material. **c** The performance of our device compares with that of several state-of-the-art photonic tensor cores[9–11,15,18,28,30] in terms of compactness, dot product operation principle, computational speed, and the available dimension of vector in a dot product. Here, the available dimension means the processor completely executes the dot product operation without the assistance of traditional digital electronic computers.

accounting for processor imperfections and environmental changes. Rapid weight updates speed up training and facilitate "on-the-fly" or online learning, which is particularly beneficial for applications such as autonomous vehicles[5].

(3) Low energy consumption and compact form factor: AI systems often deploy multiple processors (see Fig. 1a), so these processors must be energy-efficient and compact to facilitate widespread integration.

However, finding a tensor core processor that meets all these requirements simultaneously is challenging[6–14]. Traditional digital computers struggle with the speed and energy efficiency required for matrix algebra due to Joule heating, electromagnetic crosstalk, and parasitic capacitance[15,16]. In contrast, photonic integrated circuits (PICs)-based tensor core processors provide high computational speed, low energy consumption, and compactness, effectively overcoming these issues[15,17–19]. Nonetheless, developing an integrated photonic tensor core (IPTC) capable of large-scale matrix-vector multiplication with adjustable input (fan-in) and output (fan-out) sizes, alongside rapid weight updates, remains a significant challenge. For example, IPTCs utilizing wavelength-division multiplexing (WDM) are inherently limited by the number of available wavelength channels, which constrains the fan-in size in a neural network layer (see Fig. 1b)[13,20]. In addition, IPTCs based on interferometric meshes[8,18] require a single laser source but face scalability issues due to the multitude of directional couplers and phase shifters involved. To date, most IPTCs have been limited to using either static (non-volatile) weights, such as those employing phase change materials[15,21], or volatile weights based on thermo-optic effects [18], which are slow and power inefficient. These methods are unsuitable for in situ training[22]. Although some IPTC models that utilize two cascaded modulators can achieve rapid weight updates, they still require summing in the digital domain, which strongly limits the final computational speed, or they need 2N modulators for performing dot product operations on two N-dimensional vectors[23–26]. Recently, a solution first proposed by De Marinis et al.[27], which uses photocurrent integrators to perform accumulation operations in a time-division multiplexing (TDM) scheme, has been gaining attention[28,29].

Here, we introduce an IPTC with thin-film lithium niobate (TFLN) photonics and charge-integration photoreceivers (Fig. 1b), combining the advantages of photonics and analog electronics. Our processor can perform large-scale matrix-vector multiplications at high computational speeds[9–11,15,18,28,30], as quantified in Fig. 1c. This fully integrated processor, comprises only two TFLN modulators, an III−V laser, and a charge-integration photoreceiver (Fig. 1b). By adjusting the integration time of the charge-integration photoreceiver, we can flexibly modify the fan-in size for matrix-vector multiplications. Our processor can handle a fan-in size of 131,072−significantly surpassing the capacity of previously reported IPTCs by four orders of magnitude (Fig. 1c). Leveraging the high modulation speed of TFLN modulators and the fast accumulation operation of charge-integration photoreceivers[31–33], our tensor core achieves a computational speed of 120 GOPS. Moreover, with a weight update speed of 60 GHz, our tensor core enables fast in situ training. Our device successfully classifies (supervised learning) and clusters (unsupervised learning) 112 × 112-pixel images via in situ training. Notably, to the best knowledge, our device is the first to provide a solution for performing multiplications between two negative numbers, thanks to the ability of TFLN modulators to operate across a wide wavelength range. Thus, our device is the first one capable of performing in situ training for clustering images. For compactness, our tensor core employs hybrid integration techniques to combine the TFLN chip with III−V lasers and photodetectors[34,35].

## Results

### Concept and principle
Figure 1a presents a schematic of the proposed TDM-based IPTC, consisting of two cascaded TFLN Mach-Zehnder modulators, one laser, and one charge-integration photoreceiver. Our device uses charge-integration photoreceivers for accumulative operations and leverages TFLN Mach-Zehnder modulators for high-speed multiplication operations and weight updates. Therefore, using just four physical components, our IPTC can implement an entire layer of a neural network with $n$ fan-in and $m$ fan-out. $n$ and $m$ can be dynamically adjusted as needed. In contrast, the conventional WDM-based IPTC requires $n$

modulators, $n \times m$ weight additions, and $m$ large-bandwidth photodetectors to implement a layer with $n$ fan-in and $m$ fan-out (see Fig. 1b).

The input data is flattened into a vector and modulated on a time basis becoming $X(t) = \sum_{j=1}^{n} x_j \int_0^\infty \delta(t - j/f_s) \, dt$, where $n$ is the dimension of the input vector, $\delta$ is the Dirac delta function, and $f_s$ is the baud rate of the modulator. Simultaneously, the weights of the $i^{th}$ row are flattened into a vector becoming $W_i(t) = \sum_{j=1}^{n} w_{ij} \int_0^\infty \delta(t - j/f_s + \Delta T) \, dt$, where $\Delta T$ is a delay time that needs to be calibrated to guarantee each weight vector element can be correctly multiplied by the corresponding element of the input vector. At a time, $t$, $x_j$ is modulated by $w_{ij}$ performing the multiplying operation of the $j^{th}$ element. In this way, the multiplication of all of the elements of $X(t)$ and $W_i(t)$ are multiplied sequentially and then summed by the integrator. Therefore, the dot product operation between the input and weight vectors, $Y_i = \sum_{j=1}^{n} x_j w_{ij}$, can be obtained by simply reading the output voltage of the integrator through an analog-to-digital converter whose sampling rate only needs to be $f_s/n$. By adjusting the integration time of the integrator, $n$ can be changed and made to be very large. By computing the output of each node sequentially in a time series, our device can implement an entire layer of a neural network. Therefore, our processor can offer the flexibility to dynamically change the sizes of fan-in and fan-out in a layer. Moreover, through this architecture, our device can perform fast in situ training because it can update the weight vectors at the modulation speed of the modulator.

### Prototype
Figure 2a presents a photo of a prototype of our device. In addition, Fig. 2b−e provides zoomed-in micrographs of the fabricated TFLN chip, flip-chip photodetectors, traveling-wave electrodes of the modulator, and laser, respectively. More details regarding the fabrication of the TFLN chip can be found in Methods. Using flip-chip bonding technology, two photodetectors (marked as PD1 and PD2), in a balanced detection scheme, were affixed above two grating couplers, as shown in Fig. 2c. The laser and TFLN chip were connected using a photonic wire bond whose shape can be adapted to match the actual positions of the waveguide facets (see Fig. 2e). As shown in the right side of Fig. 2c, we also connected our TFLN chip with a fiber array by photonic wire bonds for calibrating bias voltages and delay time, and assisting in the multiplication involving two negative numbers. Details regarding the photonic wire bonding technology are shown in Methods and Supplementary Note 1. Figure 2f illustrates the relative heights of the TFLN chip, laser, and photodetectors.

Figure 2g presents the light-current-voltage (L-I-V) curves for the light coupled into the TFLN chip from the laser with a wavelength of 1307.22 nm. More detailed performances of the hybrid integrated laser are shown in Supplementary Note 2. Thanks to the periodic capacitively loaded traveling-wave electrodes (see Fig. 2d)[32,36,37], the 3-dB electro-optic bandwidth of our modulator is broader than 60 GHz (see Fig. 2h). The output voltage of the integrator linearly increases with the integration time for a constant input optical power (see Fig. 2i). In a balanced detection scheme, when the optical power received by PD1 is lower than that received by PD2, the output voltage variation of the integrator is positive and, when it is higher than that received by PD2, the output voltage variation of the integrator is negative. This means that the proposed photoreceiver can perform add and subtract operations in the matrix-vector multiplication. More details regarding the charge integrator and the corresponding electrical controlling circuit can be found in Supplementary Note 3.

### Dot product accelerator
In this section, we demonstrate how to perform a dot product operation between two vectors using our device. A schematic of data flows through our device is shown in Fig. 3a. Python, an open-source programming language, was used to control all our devices. We recorded 3780 photonic dot product measurements using our device

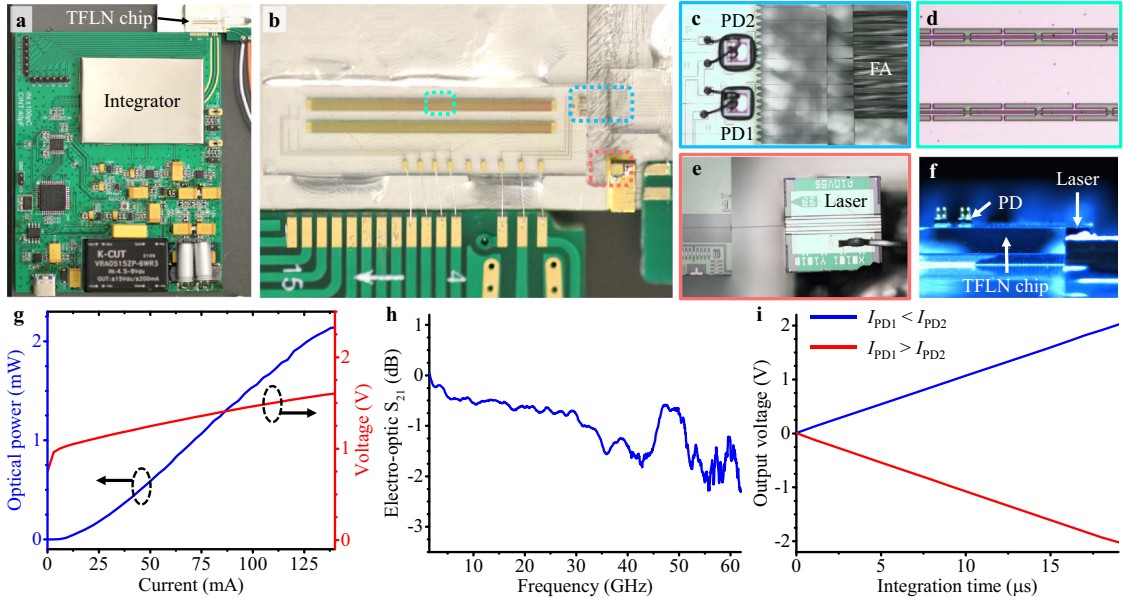

**Fig. 2 | A prototype of the packaged device. a** Photo of the entire device. The top is our hybrid integrated chip; the Bottom is the electric control and power supply circuits of the charge-integration photoreceiver. **b** is a micrograph of our hybrid integrated chip, including the fabricated TFLN photonic circuit, flip-chip photo-detectors, and laser. **c–e** is the zoomed-in micrographs of flip-chip photo-detectors (PDs), the traveling-wave electrode of the modulator, and the laser, respectively. **f** A micrograph of the side view of our device, showing the relative heights of the TFLN chip, laser, and photodetectors. **g** The light-current-voltage

curves for the light coupled into the TFLN chip from the laser. **h** Electro-optic bandwidth ($S_{21}$ parameter) of the modulator. **i** The output voltage of the photo-receiver varies with the integration time when the input optical power is fixed at a certain value. In a balanced detection scheme, when the optical power received by PD1 is lower than that received by PD2, the output voltage variation of the inte-grator is positive, and, when it is higher than that received by PD2, the output voltage variation of the integrator is negative.

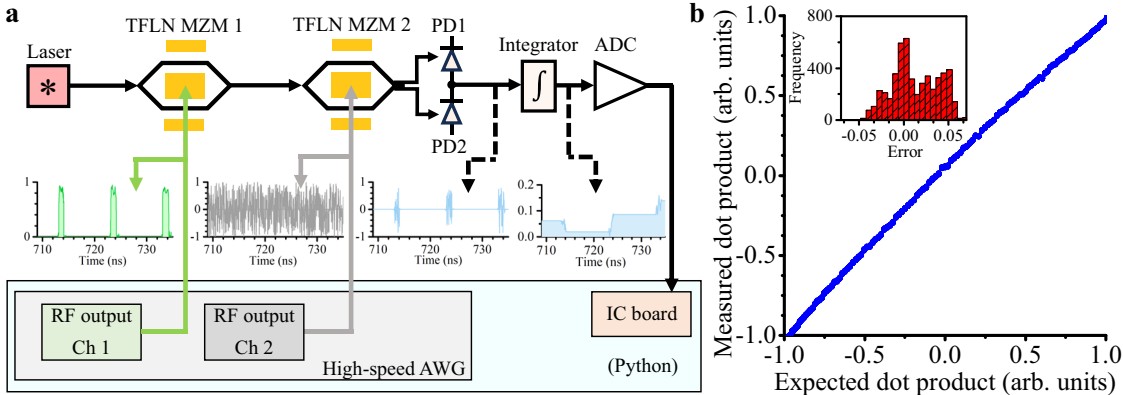

**Fig. 3 | Experimental result for dot product operation with our device. a** A schematic of the working principle of our device. The light is emitted from a laser and then passes through two cascaded thin-film lithium niobate (TFLN) Mach-Zehnder modulators (MZMs). The TFLN MZMs are driven by a high-speed arbitrary waveform generator (AWG). The light is then received by two photodetectors (PDs, marked as PD1 and PD2) in a balanced detection scheme, and the corresponding photogenerated electrons are accumulated in the integrator. Reading the output

voltage of the integrator with ADC, we can obtain the dot product result. ADC: analog-to-digital converter. IC: integrated circuit. DAC: digital-to-analog converter. **b** The results of dot product operation between two 131072-dimensional vectors performed by our device with a computational speed of 120 GOPS. Compared with the expected dot product results, the error of the measured ones has a standard deviation of 0.03 (6.04 bits).

by randomly varying the two vectors. The dimension of each vector was set at 131072, a limit imposed by our high-speed arbitrary wave-form generator (AWG). The two vectors were modulated separately by two modulators at a modulation rate of 60 Gbaud, enabling a com-putational speed of 120 GOPS, and a weight update speed of 60 GHz. The time delay between two vectors was initially calibrated to guar-antee that each element of the first vector can be correctly multiplied by the corresponding element of the second vector. More details regarding the experimental setup can be found in Supplementary Note 4. The measured output voltage (*i.e.*, dot product result), scaled

between −1 and +1, as a function of the expected dot product result, is shown in Fig. 3b. Compared with the expected dot product result, the error of the measured one has a standard deviation of 0.03 (6.04 bits)—more than the 4 bits of precision required for performing AI tasks (details can be found in Supplementary Note 4)[38].

## Images classification

We built a multilayer perceptron (see Fig. 4a) and tested it against the MNIST large-scale handwritten digit database[39,40]. Here, the multilayer perceptron includes 4 layers: an input layer, two hidden layers, and an

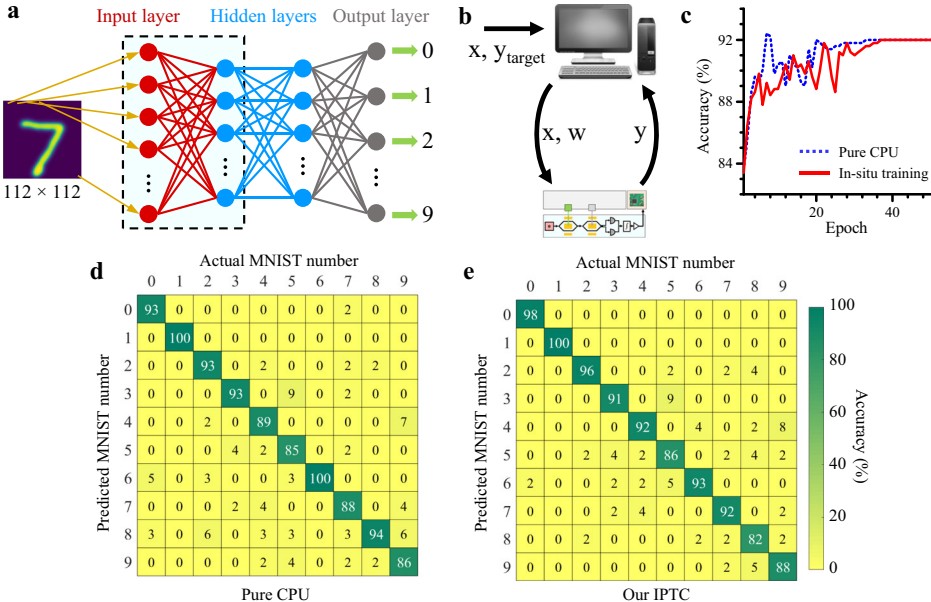

**Fig. 4 | Classification results of handwritten digits using our device. a** A block diagram of a multilayer perceptron neural network, which consists of an input layer, two hidden layers, and an output layer that provides classification outputs. **b** A schematic of the in situ training, a form of online training, where our IPTC handles forward propagation while the computer manages the nonlinearity function and backpropagation. **c** The validation accuracy as a function of epoch for in situ training (solid red line) scheme compared to that running on just a central processing unit (CPU, dashed blue line). **d**, **e** Theoretically calculated confusion matrices (purely run by the CPU) and experimental confusion matrices (run by our IPTC) using the MNIST large-scale database[39]. For "in situ" training, 2000 handwritten digits are used for training, and 500 digits are used for testing. Our IPTC achieves classification accuracy comparable to that achieved by the CPU.

output layer. Each handwritten digit image, having 112 × 112 pixels, was flattened into a 12544 × 1 vector as the input of the first layer. The number of nodes in the first and second hidden layers was set to 70 and 300, respectively, and the leaky ReLU function was used for the nonlinear activation function[41].

Classification is a supervised learning AI task that requires labeled data to train the model. Our multilayer perceptron model was trained with 2000 labeled digit images using an in situ training scheme (see Fig. 4b) that our IPTC performs forward propagation. At the same time, the electronic computer handles nonlinearity function and backpropagation. Weight vectors were updated by the stochastic gradient descent method[42], allowing individual samples to be trained iteratively. The training process from forward propagation to backpropagation was repeated until convergence. Figure 4c shows the validation accuracy as a function of epoch for in situ training scheme compared to that running on just a central processing unit (CPU). More details regarding the training algorithm and the interfaces between the central processing unit (CPU) and the optoelectronic assembly can be found in the Methods and Supplementary Note 4.

The confusion matrix for 500 images (Fig. 4d, e) shows an accuracy of 91.8 % for the generated predictions, in contrast to 92% for the numerical results calculated on a CPU. Our IPTC achieved near theoretical accuracy, indicating that the in situ training scheme enables the system to inherently account for the hardware nonidealities, including fabrication variations and noise. Essentially, the nonidealities are "baked into" the training process. This has also been experimentally demonstrated in ref. 3.

**Images clustering**

Supervised learning can successfully solve real-world challenges, but it has some drawbacks. One of the main limitations is that it requires a large number of accurately labeled data to train the model[43,44]. Creating such a database is a time-consuming and resource-intensive task that may not always be feasible. In contrast, unsupervised learning can be operated on unlabeled data to discover its

underlying structure, offering an alternative approach for extracting data features.

We demonstrate the potential of our device for unsupervised learning AI tasks by utilizing it to cluster the MNIST large-scale handwritten digits with principle component analysis − one of the most commonly used unsupervised learning models[45]. Principle component analysis simplifies high-dimensional data by geometrically projecting them onto a limited number of principal components (PCs), *i.e.*, unit vectors, to obtain the best summary of the data[45]. Clustering handwritten digits with principle component analysis involves two main steps: (1) finding the PCs for the unlabeled database, *i.e.*, training the model, and (2) projecting the data onto each PC. Here, we used the power method to find the PCs that[46]

$$\mathbf{b}_{i+1} = \frac{\mathbf{A}\mathbf{b}_i}{\| \mathbf{A}\mathbf{b}_i \|}, \tag{1}$$

where $\mathbf{A} = \mathbf{X}^T\mathbf{X}$, $\mathbf{X}^T$ means the transpose of $\mathbf{X}$, $\mathbf{X}$ is a $p \times n$ data matrix with column-wise zero empirical mean, and $p$ and $n$ are the total number of handwritten digits and the pixels of each digit, respectively. $\mathbf{b}_i$ is a $n \times 1$ unit vector, obtained at the $i^{th}$ iteration, and $\mathbf{b}_0$ is a randomly generated unit vector. $\mathbf{b}_i$ converges to the first PC (PC1) when the variance of the projected points, $\mathbf{X}\mathbf{b}_i$, achieves the maximum value. The subsequent PCs can be obtained by a similar approach after subtracting all the previous PCs from $\mathbf{X}$. More details regarding the power method can be found in Methods.

Through Eq. (1), we can know that the training involves the multiplication between two negative numbers. To achieve this, as illustrated in Fig. 5a, we found a solution that injects light with a wavelength of $\lambda_2$ into the second modulator, in addition to injecting light with a wavelength of $\lambda_1$ into the first modulator. When the phase difference between the two arms of the first and second modulators is adjusted to $\theta_1$ and $\theta_2$, respectively, the output of the balanced photodetectors becomes $I_0 \sin\theta_1 \sin\theta_2$, indicating that this method enables the multiplication between two numbers with any signs. More

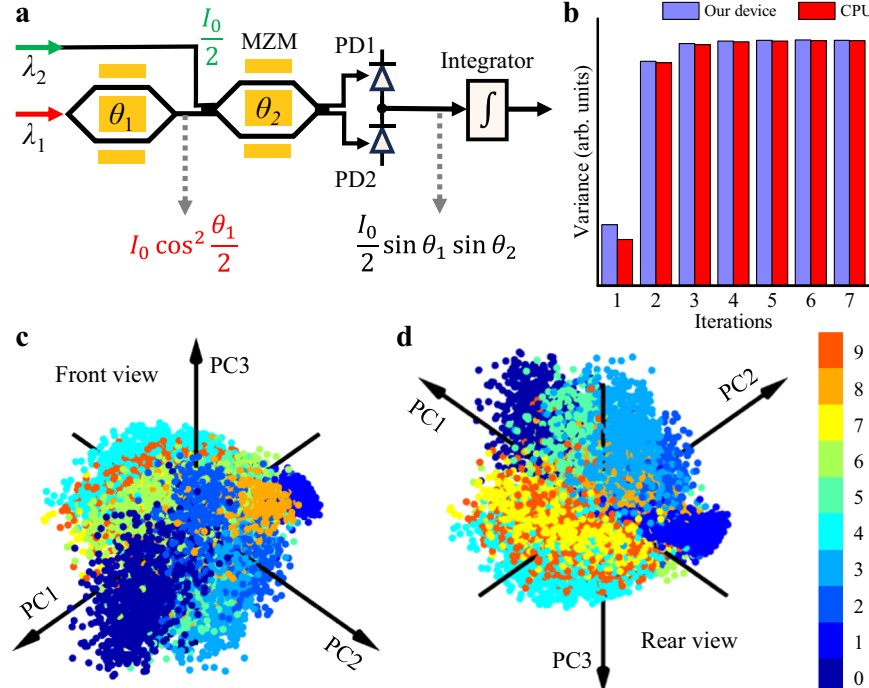

**Fig. 5 | Clustering result of the handwritten digits using our device. a** A schematic of the working principle of our device to perform the multiplication between two numbers with any signs, including two negative numbers. MZM: Mach-Zehnder modulator. PD: photodetector. **b** The variance of the projected points, $\mathbf{Xb}_i$, as a function of iterations when finding the first principle component using the power method. $\mathbf{X}$ is a $p \times n$ data matrix, representing the 10000 of the handwritten digits from the MNIST large-scale database[39]. $\mathbf{b}_i$ is a $n \times 1$ unit vector, obtained at the $i^{th}$ iteration. The algorithm performed by our device has a comparable iteration speed with that of the central processing unit (CPU). **c, d** are the front and rear views of the 3D coordinates of each handwritten digit based on the scores of projecting onto the first three principal components (PCs), respectively.

details regarding the working principle can be found in Supplementary Note 4. The convergence speed of our device is comparable with that of the CPU (see Fig. 5b), demonstrating that a dot product computing precision of 6.04 bits is enough to train the model using the power method.

Using the power method discussed above, we can obtain all the principal components. However, in general, the first few PCs are sufficient to summarize the data. In our example, the projections onto PC1-PC44 encompass 90% of the features. To visualize the clustering result of the handwritten digits using our device, Fig. 5c and d present the projections onto PC1-PC3, accounting for 28.7% of the features. Although only the first three PCs are used, the unlabeled handwritten digits can still be well clustered. Moreover, projecting the data onto the three PCs using our device is 5 times faster than a CPU (Intel i9-9900 @ 3.10 GHz).

### TDM and WDM-based architecture

To show the scalability of our solution, we propose an end-to-end photonic neural network that combines the benefits of TDM and WDM methods, as illustrated in Fig. 6. This network is capable of executing multiple AI tasks simultaneously, spanning from the input to the output layer, with nanoseconds latency, all without relying on a digital processor for assistance. As an example, shown in Fig. 6, is a proposed neural network that includes 4 layers: an input layer, two hidden layers, and an output layer.

(1) From the input layer to the hidden layer 1. The information of $K$ AI tasks is encoded by $K$ input TFLN modulators and transmitted on $K$ corresponding wavelengths. These signals from input TFLN modulators are then split and channeled into $m$-weighted TFLN modulators. Although some waveguides must pass through $(K-1)$ crossings, the total insertion losses remain manageable, as an insertion loss of 0.02 dB per crossing has been demonstrated[47]. Following this,

each weighted TFLN modulator feeds into a $K$-channel WDM, which separates the wavelengths to $K$ charge-integration photoreceivers for generating vector-vector dot products. $K$ commercial complementary metal-oxide-semiconductor (CMOS) switches[48] control the output timings of these photoreceivers. In addition, a CMOS comparator, which selects the maximum between the input and reference voltages, facilitates the ReLU activation function of each vector vector dot product[49]. The use of 90 wavelengths from a comb source for photonic neural networks[30] and a 64-channel integrated WDM[50] have been previously demonstrated, making $K$, $m = 64$ a practical choice. With this setup, we can achieve a computational speed of 491 TOPS and an energy efficiency of 6.5 fJ/OP (*i.e.*, 153 TOPS/W), factoring in a modulation speed of 60 Gbaud/s, including the energy consumption of the laser, DACs, charge-integration photoreceivers, CMOS switches, and CMOS comparators. Further details are available in Supplementary Note 6.

(2) From hidden layer 1 to 2, and from hidden layer 2 to the output layer, conventional WDM-based architectures[3,15] are employed. These parts are unaffected by limitations related to fan-in size, thanks to the relatively small numbers of nodes in the hidden layers. The outputs of hidden layer 1 are fully connected to the $m$ neurons in hidden layer 2. Similarly, the $h$ outputs from hidden layer 2 are fully connected to the $p$ neurons of the output layer, resulting in $p$ network outputs.

This hybrid processor enables the sequential processing of 64 images, each with a resolution of $112 \times 112$ pixels, within 62.5 ns. Its significant potential extends to various fields, including autonomous vehicles requiring simultaneous image processing from multiple cameras.

### Discussion

Our device's computational speed, compactness, and ability for large-scale dot product operations are summarized in Fig. 1c, where the

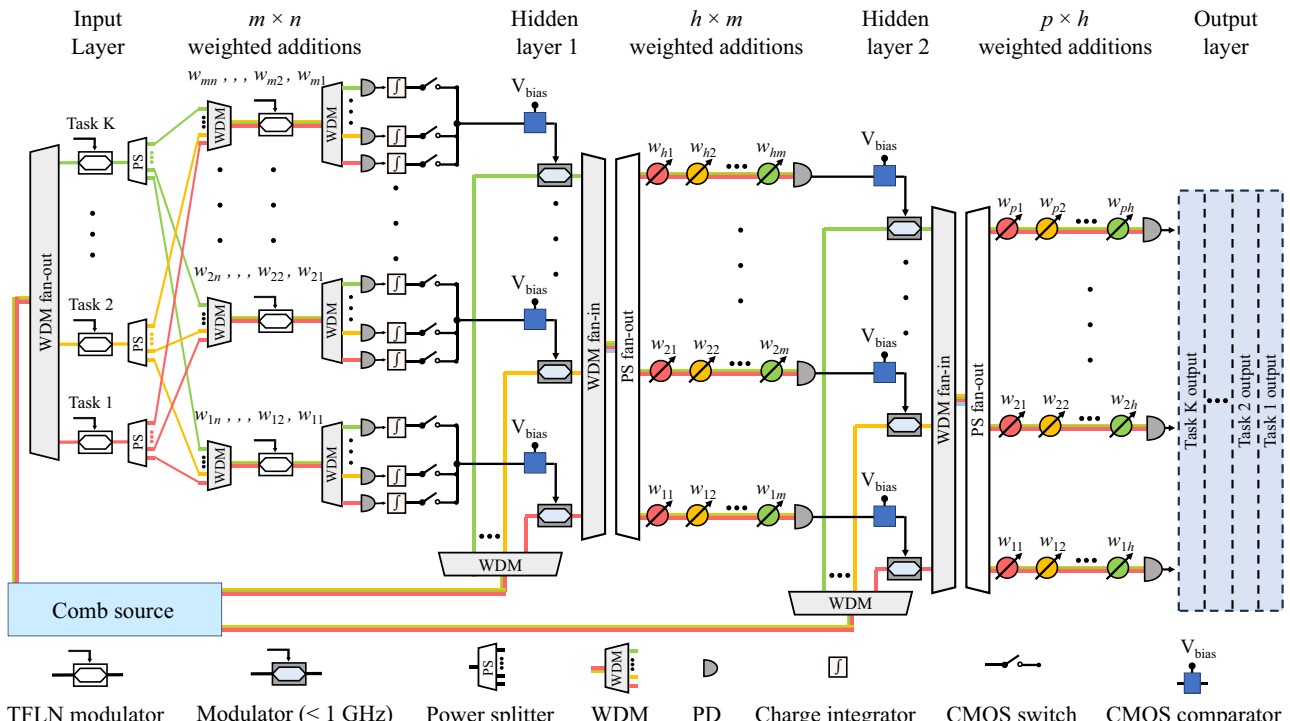

**Fig. 6 | A schematic of a photonic neural network designed to show the scalability of the proposed integrated photonic tensor core, employing a hybrid approach that combines time-division multiplexing (TDM) and wavelength-division multiplexing (WDM).** This photonic neural network is capable of processing multiple tasks in parallel. As an example, the proposed network includes four layers. Matrix multiplication between the input layer and hidden layer 1 is performed using the TDM method. Subsequent matrix multiplications--from hidden layer 1 to hidden layer 2, and from hidden layer 2 to the output layer--are carried out using the WDM method. A comb source generates multiple wavelengths to facilitate these operations. Complementary metal-oxide-semiconductor (CMOS) comparators are utilized to implement activation functions. TFLN: thin-film lithium niobate. PD: photodetector.

performance of our device is compared to that of other state-of-the-art photonic devices[9–11,15,18,28,30]. Our device can simultaneously achieve high performance across all these aspects. Note that "ability" refers to our processor to perform dot product operations without the assistance of a digital processor for accumulation operations. According to the demonstrated performance of the TFLN modulator[51], our IPTC can achieve a computational speed of over 520 GOPS, and the vector dimension can extend to over one million (the dotted triangle shown in Fig. 1c) using more advanced modulator driver boasting higher speeds and larger memory capacity.

Compared to photonic processors based on free-space[52] or discrete[30] optics, our fully integrated processor is compact. However, the footprint of fast modulators is not the primary reason for choosing the TFLN technology. The size of high-speed drivers, often the bottleneck in achieving an ultra-high compute density, must also be considered. Our choice of TFLN modulators is driven by their ability to: (1) simultaneously achieve a low insertion loss, CMOS-compatible voltage, and a broad electro-optic bandwidth[32]. (2) Operate across a wide wavelength range, perform multiplication of two negative numbers, and for their compatibility with hybrid TDM and WDM architectures, as previously discussed, and (3) exhibit no modulation loss, unlike silicon modulators, which suffer from variable insertion loss depending on the applied voltage.

In summary, we have experimentally demonstrated that our device can perform large-scale matrix-vector multiplications with flexibly adjustable fan-in and fan-out sizes and facilitate rapid weight updates. Our device is the pioneering IPTC with the capability to handle the multiplication between two negative numbers. It is capable of processing both supervised and unsupervised learning AI tasks through in situ training. Thanks to its compatibility with current commercial optical transceivers, our solution has the potential to rapidly enter the commercial phase. By taking advantage of electronic and photonic analog computing, our research paves the way for developing a universal IPTC.

## Methods
### Design and fabrication of TFLN chip
The proposed TFLN chip was fabricated using a wafer (NanoLN) consisting of a 360 nm thick, x-cut, y-propagating, LN thin film on a 500 μm thick quartz handle with a 2 μm SiO₂ layer in between the two. The optical devices were patterned using optical lithography and etched using inductively coupled plasma. Then, a cladding layer with a 1 μm thick SiO₂ was deposited on the top of optical devices. Gold and heater electrodes were then patterned with a lift-off process. To achieve a low-voltage, high-bandwidth electro-optic solution, we used 1 cm-long, capacitively loaded, traveling-wave electrodes on our TFLN modulators. Our modulators exhibit a 3-dB electro-optic bandwidth broader than 67 GHz, a $V_\pi$ of 2.4 V, and an extinction ratio larger than 20 dB. For the TDM and WDM-based architecture, We note that during the review process of this paper, other teams were developing similar schemes, which constitute the first part of the photonic neural network shown in Fig. 6, using bulk fiber-optic modulators, with results presented in conference proceedings[25,26].

### Photonic wire bonding process
Photonic wire bonding is a technique for building hybrid connections between disparate optical components, such as TFLN chips, lasers, and fiber arrays, using three-dimensional (3D) polymer waveguides created by in situ, two-photon polymerization[53]. In our case, the photonic wire bonds were used between the TFLN chip and the laser. We also used the photonic wire bonds in various locations of our hybrid photonic circuit for calibration and testing purposes. The hybridization process

was carried out as follows: first, the TFLN chip, a fiber array (used in calibration and testing), and the laser were glued to an aluminum submount with a low alignment precision using ultraviolet curable epoxies; second, the photoresist was dispensed on the optical ports of the TFLN chip, fiber array, and laser; third, the photonic wire bonder (Vanguard Automation GmbH, SONATA1000) was used to expose the photoresist and develop the shape of the interconnecting photonic wires. More detailed performances of photonic wire bonds and the hybrid integrated laser are shown in Supplementary Note 2.

### Experimental setup

A vector network analyzer (Agilent N5227A) with a bandwidth of up to 67 GHz was used to characterize the electro-optic response of the fabricated modulator at a telecom wavelength of 1310 nm. To perform the dot product operation, our device was driven by an arbitrary wave generator (Keysight, M8194A). For comparison purposes, the machine learning algorithms are also executed using a CPU (Intel i9-9900 @ 3.10GHz). More details can be found in Supplementary Note 4.

### The principle of the stochastic gradient descent method

For the multilayer perceptron, the weight vectors in this study were updated using the stochastic gradient descent method, allowing individual samples to be trained iteratively. The training was implemented using a labeled dataset $(\mathbf{x}, \mathbf{t})$, where $\mathbf{x}$ is the network input, and $\mathbf{t}$ is the target to be compared with the network output. In the forward propagation, the output vector, $\mathbf{z}^{(l)}$, of the $l^{th}$ layer can be given by

$$\mathbf{z}^{(l)} = \mathbf{w}^{(l)} \cdot g(\mathbf{z}^{(l-1)}), \tag{2}$$

where $\mathbf{w}^{(l)}$ represents the weight matrix between the $(l-1)^{th}$ and $(l)^{th}$ layers, $g(\mathbf{z}^{(l-1)})$ is the activation function for the output of the $((l-1)^{th}$ layer, and $\mathbf{z}^{(1)} = \mathbf{x}$.

Through backpropagation, the "error", $\boldsymbol{\delta}^{(l-1)}$, of the $(l-1)^{th}$ layer can be calculated by

$$\boldsymbol{\delta}^{(l)} = (\mathbf{w}^{(l)})^{\mathrm{T}} \cdot \boldsymbol{\delta}^{(l+1)}, \tag{3}$$

where $(\mathbf{w}^{(l)})^T$ means the transpose of $\mathbf{w}^{(l)}$, and $\boldsymbol{\delta}^{(4)} = (\mathbf{t} - \mathbf{z}^{(4)})$ in the case of our network only has 4 layers. Then, the weight matrix can be updated by

$$\mathbf{w}^{(l)} = \mathbf{w}^{(l)} + \Delta\mathbf{w}^{(l)}, \tag{4}$$

where $\Delta\mathbf{w}^{(l)} = \gamma(\boldsymbol{\delta}^{(l+1)} \odot g(\mathbf{z}^{(l+1)})) \cdot (\mathbf{z}^{(l)})^{\mathrm{T}}$, $\gamma$ is the learning rate, and $\odot$ is the Hadamard product (element-wise multiplication operator).

In our "in situ" training scheme, our IPTC performed forward propagation while the computer handled nonlinearity function and backpropagation. This training process from forward propagation to backpropagation was repeated until convergence or all samples were trained.

### The principle of power method for finding all principle components

We can find each PC by repeating Eq. (1), but the matrix $\mathbf{A}$ needs to be changed for different PCs. To find the $k^{th}$ PC, $\mathbf{w}_{(k)}$, the matrix $\mathbf{A}$ can be given by

$$\mathbf{A} = \hat{\mathbf{X}}_k^{\mathrm{T}} \hat{\mathbf{X}}_k, \tag{5}$$

where

$$\hat{\mathbf{X}}_k = \mathbf{X} - \sum_{s=1}^{k-1} \mathbf{X}\mathbf{w}_{(s)}\mathbf{w}_{(s)}^{\mathrm{T}} \tag{6}$$

$\mathbf{X}$ is a $p \times n$ data matrix with column-wise zero empirical mean, $p$ and $n$ are the number of samples and the pixels of each digit image, respectively. $\mathbf{w}_{(s)}$ means the $s^{th}$ PC.

## Data availability

The authors declare that the data supporting the findings of this study are available within the paper and its Supplementary Information files. Source data can be found at https://doi.org/10.6084/m9.figshare.26965324.

## Code availability

The code used in this study is available from the corresponding authors upon reasonable request.

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

## Acknowledgements

X.C. acknowledges the Natural Science Foundation of China (Grant No. 62293523). L.C. the SiEPICfab consortium, the B.C. Knowledge Development Fund (BCKDF), the Canada Foundation for Innovation (CFI), the Refined Manufacturing Acceleration Process (REMAP) network, the Canada First Research Excellence Fund (CFREF), and the Quantum Materials and Future Technologies Program. Z.L. acknowledges the National Natural Science Funds of China (Grant No. 62405382). The authors thank Simon Levasseur and Nathalie Bacon from the University of Laval for their technical support. The authors thank Xin Xin for his help in the experiment setup. We thank Zhongguan Lin for his help in designing the electrical control circuits. The authors thank Xiaogang Qiang for his suggestion for Fig. 1. Part of the images shown in Fig. 1 were designed by Freepik.

## Author contributions

Z.L., X.C., and L.C. jointly conceived the idea. Z.L. designed the device with help from Y.L., M.X., and T.W. M.R., G.P., J.S., P.B., and W.J. designed and fabricated DFB-buried heterostructure lasers with spot-size converters. Y.Z. and W.K. fabricated the photonic chip. Z.L., S.X.Y., M.M., and J.S. performed the photonic wire bonding experiment. Z.L. assembled the entire device. Z.L. performed the experiments with help from J.S., A.S., M.H., Z.Z., O.E., M.A.Q., and X.G. W.S., L.A.R., W.C., H.M., S.S., B.S., X.Q., N.J., and S.Y.Y. assisted with the theory and algorithm. Z.L., X.C., B.S., and L.C. supervised and coordinated the work. Z.L. and X.C., with help from B.S., wrote the manuscript with contributions from all co-authors.

## Competing interests

The authors declare no competing interests.
