## [Peer Review File · Nature Communications]

120 GOPS Photonic Tensor Core in Thin-film Lithium Niobate for Inference and in-situ TrainingREVIEWER COMMENTS

Reviewer #1 (Remarks to the Author):

5 GOPS/neuron Photonic Tensor Core with
Thin-film Lithium Niobate Photonics

The authors show an integrated processor comprising of two concatenated thin film lithium niobate Mach-Zehnder modulators, a photon source and charge integration photodetectors to implement a neural network.

The manuscript is generally too crisp and dense for the article to reach out to a wider audience: The introduction and generally the whole manuscript should be expanded. The authors may wish to explain why lithium niobate is necessary, e.g. nonlinearity.

Other elaborations needed are:

How many TFLN Mach-Zehnder interferometers are needed in the experiment for the neural network, especially in the deep photonic neural network? What is the total weights used in the full network? Is there a reason for setting the hidden layers to 70 and 300 for the image classification?

Overall, I feel that the manuscript has demonstrated an interesting result and the manuscript deserves to be accepted for publication. However, as I have pointed out, the manuscript needs substantial modification and elaboration.

Reviewer #2 (Remarks to the Author):

This manuscript presents an integrated dot product engine that is based on an optoelectronic prototype. It comprises two cascaded TFLN modulators, one hybridly assembled laser connected via a photonic wirebond and two flip-chip photodiodes, using an externally connected electronic assembly circuit for integrating the received signal over time and providing the accumulation operation within the neural network layout. This device is used for demonstrating i) MNIST dataset classification, employing also “hardware-in-the-loop” training, and ii) PCA in the MNIST dataset. Although the prototype is really impressive and its use within a “hardware-in-the-loop” training process is innovative, I cannot recommend its publication at Nature Communications for reasons explained below:

1. The most important reason has to do with the bold claims about the speed, compactness and capacity advantages of their device against state-of-the-art photonic neural network deployments. There are several sections in the paper where these advantages are highlighted, including Introduction (“Our processor significantly surpasses current photonic platforms [9–11, 15, 18, 23, 25], as quantified in Fig. 1c.”), saying then that “our processor achieves a computational speed of 65 GOPS per neuron, including concurrent weight updates—a speed hitherto unachieved, enabling fast “hardware-in-the-loop” training”), and Discussion-Conclusion (“where the performance of our device is compared to that of other state-of-the-art photonic devices [9–11, 15, 18, 23, 25]. Our device excels in all these aspects.”). Fig. 1(c) provides also a pictorial representation of the computational speed versus dimension of dot product for so far demonstrated photonic neural network prototypes. However:

a. regarding speed, this is not the highest speed that is reported in photonic neural network deployment. The authors claim in their introductory paragraph that “To date, all IPTC models are confined to either static (non-volatile) weights, like those utilizing phase change materials [15, 21], or slowly updated (volatile) weights through thermal control [18]”, but the work presented by Giamougiannis et al in Giamougiannis G. et al, "Analog nanophotonic computing going practical: silicon photonic deep learning engines for tiled optical matrix multiplication with dynamic precision" *Nanophotonics*, vol. 12, no. 5, 2023, pp. 963-973. <https://doi.org/10.1515/nanoph-2022-0423> and Giamougiannis G. et al, "Neuromorphic silicon photonics with 50 GHz Tiled Matrix Multiplication for Deep Learning applications", *SPIE Advanced Photonics*, vol. 5, no. 1, 016004, 2023 employs input and weight updates at 50GHz baudrates, i.e. 100TOPS, for executing the linear part of multiple neural layers in the optical domain,

b. regarding compactness, the TFLN phase section length is already 1cm, making it hard to imagine how this could form the most compact solution. long and this claim is completely unjustifiable based on the metrics provided in the manuscript. Authors should calculate the area efficiency of their TFLN-based modulation scheme in GOPS/mm² as a metric and compare this to the respective area efficiencies of alternative modulator technologies that could be employed in photonic neural network experiments (i.e. SiGe EAMs, InP MZMs, Silicon Microring Modulators etc) in order to back-up their claim about compactness.

c. regarding capacity, this is definitely not a property of the device prototype but of the memory supported by the commercial DACs employed in the experiment.

Based on the above, it is in general not clear what are the advantages of using TFLN modulators instead of alternative integrated modulator technology and what are the main advances that this device brings in the field of photonic neural networks.

2. Probably the most innovative part in this work has to do with the use of the optoelectronic prototype in the “hardware-in-the-loop” training process. However, more information is needed in order to understand all the details of this demonstration. The authors should provide details about the interfaces employed between the CPU and the optoelectronic assembly, how was the model initialized and how were the normalization stages implemented, how the digital weight update values obtained from the CPU were translated into analog voltages, how many epochs were required until converging creating a graph that shows the convergence over time for both the purely digital and the “hardware-in-the-loop” training process would be helpful), how did the training model deal with the noise of the optoelectronic system, how does this solution compare to the training of the respective model exclusively in the digital domain, what was the bit resolution per neural layer required when training in the digital domain etc. A graph showing how the performance of the training process varies when the bit resolution of every individual layer drops below the ideal value would be very helpful.

3. Authors should provide more details on how the energy efficiency of 800fJ per operation has been calculated and what this includes. Are DACs and drivers taken into account, as this should be the case in order to allow for a fair comparison with alternative neural network hardware deployment? A detailed breakdown of the energy consumption should be provided. Moreover, one of the main motivations for photonic neural network developments is their potential to drive energy efficiency to values lower than current state-of-the-art, well below sub-pJ/MAC. Can the proposed scheme scale towards supporting competitive energy efficiency values and, if yes, how?

4. The Introduction section refers at the end of its second paragraph: “...Additionally, IPTCs based on interferometric meshes [8, 18] require a single laser source but face scalability issues due to the multitude of directional couplers and phase shifters involved. To date, all IPTC models are confined to either static (non-volatile) weights, like those utilizing phase change materials [15, 21], or slowly updated (volatile) weights through thermal control [18], which are power intensive, rendering them unsuitable for “hardware-in-the-loop” training [22]. While these platforms can achieve high computational speeds, they are limited to small-scale vector operations and inference tasks.” The state-of-the-art on interferometric meshes should be enriched to include developments and architectures demonstrated by the groups of O. Liboiron-Ladouceur, N. Pleros and N. Youngblood. Moreover, additional important work regarding the TDM scheme should be referenced, like the work of L. DeMarinis et al., “A Codesigned Integrated Photonic Electronic Neuron,” in IEEE Journal of Quantum Electronics, vol. 58, no. 5, pp. 1-10, Oct. 2022, Art no. 8100210, doi: 10.1109/JQE.2022.3177793 and Z. Zhong et al, “Lightning: A Reconfigurable Photonic-Electronic SmartNIC for Fast and Energy-Efficient Inference”, in Proc. of the ACM SIGCOMM 2023 Conference (ACM SIGCOMM '23) New York, NY, USA, 452–472, <https://doi.org/10.1145/3603269.3604821> .

5. The Discussion section presents a conceptual layout of a possible architecture that combines TDM and WDM. It is a very high-level setup that exploits techniques that have not been validated so far as such and, for that reason, it is rather questionable on whether this comprises indeed a valid perspective. For example, the weighting stage at the Input Layer exploits TFLN modulators for weighting multiple input channels simultaneously, which has not been demonstrated experimentally and raises several concerns regarding its feasibility and performance. Is it possible in this case, for example, to apply again the concept for two negative number multiplication? On the same line, the rather simplistic assumption of utilizing 90 wavelengths from a single comb source towards reaching 500TOPs builds upon i) a broadband operational regime for the TFLN modulators, ii) a high amount

of 90 cascaded resonant devices in hidden layer #2 and in the output layer, and iii) 90-ch MUX/DEMUX elements. Moreover, there are several waveguide crossings and delays foreseen in this layout, which are difficult to envision in integrated form. The total losses of the layout and the noise are also not discussed at all. This section should be rewritten so as to extend along a more scientifically and technologically validated roadmap, including either simulation results or at least details about every of its stages and components, or alternatively should be omitted in order to avoid misleading conclusions.

Type: Research article

Title: 120 GOPS/neuron Photonic Tensor Core with Thin-film Lithium Niobate Photonics

Dear Reviewers,

We want to thank you for your time in reviewing our manuscript (NCOMMS-23-57567). We found the reviewers' report very insightful. In response, we have carefully revised our paper in accordance with the comments provided by the editor and reviewers. Enclosed, please find our detailed point-by-point response to each comment. We believe that our manuscript has been significantly strengthened as a result.

Specifically, to improve the quality and impact of our revised manuscript, we have implemented the following key changes based on the reviewers' suggestions:

- 1) We have conducted new experiments to demonstrate that our processor can achieve a computation speed of 120 GOPS. Please note that the originally submitted manuscript had reported a computation speed of 65 GOPS.
- 2) We have included results on the convergence as a function of time for both the purely digital and the "hardware-in-the-loop" training.
- 3) We have added results regarding the validation accuracy on the MNIST dataset as a function of the effective resolution.
- 4) We have incorporated discussions on the advantages of TFLN modulation in photonic neural networks.
- 5) We have added a new section to provide additional details regarding Fig. 6 (the scalability of the proposed integrated photonic tensor core).
- 6) We have added a table comparing the compute density of processors using different modulators.
- 7) We have added two tables to illustrate the energy efficiency of our processors.
- 8) We have provided more details about the interfaces employed between the CPU and the optoelectronic assembly.

Below is our point-by-point response to the referees' reports. For clarity, the reviewers' comments are presented in **blue text**, our responses in black, and quotations from our revisions (including all figures from our revisions) in *black italic text*

Reviewer #1

Comment 1: The authors show an integrated processor comprising of two concatenated thin film lithium niobate Mach-Zehnder modulators, a photon source and charge integration photodetectors to implement a neural network.

The manuscript is generally too crisp and dense for the article to reach out to a wider audience: The introduction and generally the whole manuscript should be expanded. The authors may wish to explain why lithium niobate is necessary, e.g. nonlinearity.

Response: Thank you for your comments. According to the reviewer's suggestion, we have rewritten some sentences and added why lithium niobate is necessary.

Line 6 to Line 17, Paragraph 2, Page 13, main manuscript:

“Compared to photonic processors based on free-space [53] or discrete [29] optics, our fully integrated processor is compact. However, the footprint of fast modulators is not the primary reason for choosing the TFLN technology. The size of high-speed drivers, often the bottleneck in achieving an ultra-high compute density, must also be considered. Our choice of TFLN modulators is driven by their ability to: 1) simultaneously achieve a low insertion loss, CMOS-compatible voltage, and a broad electro-optic bandwidth [31]; 2) Operate across a wide wavelength range, perform multiplication of two negative numbers, and for their compatibility with hybrid TDM and WDM architectures, as previously discussed; and 3) exhibit no modulation loss, unlike silicon modulators, which suffer from variable insertion loss depending on the applied voltage.”

Line 34 to Line 38, Paragraph 3, Page 3, main manuscript:

“By adjusting the integration time of the charge-integration photoreceiver, we can dynamically modify the fan-in size for matrix-vector multiplications. Our processor can handle a fan-in size of 131,072—significantly surpassing the capacity of previously reported IPTCs by four orders of magnitude.”

Line 1 to Line 4, Paragraph 1, Page 4, main manuscript:

“Notably, to the best knowledge, our device is the first to provide a solution for multiplications involving two negative numbers, thanks to the ability of TFLN modulators to operate across a wide wavelength range.”

Comment 2: Other elaborations needed are: How many TFLN Mach-Zehnder interferometers are needed in the experiment for the neural network, especially in the deep photonic neural network? What is the total weights used in the full network? Is there a reason for setting the hidden layers to 70 and 300 for the image classification?

Response: Thank you for your question.

1) Only two TFLN Mach-Zehnder modulators are required to implement a single layer of a neural network in the experiment. Using a charge-integration photoreceiver, our device can perform dot product operations for vectors of large dimensions (over 100,000) without the assistance of a digital processor for accumulation operations. However, employing more TFLN Mach-Zehnder modulators can lead to higher computation speeds. For example, as illustrated in Fig. 6 of the revised manuscript, we can potentially achieve a computation speed of 491 TOPS.

2) For image classification involving 112×112 -pixel images, the number of total weights is $70 \times 12,544$.

3) The number of hidden layers can vary. Here, we set the hidden layers to 70 and 300 as these values were sufficient for convergence.

Fig. 6, main manuscript:

Fig. 6 A schematic of a photonic neural network designed to show the scalability of the proposed integrated photonic tensor core, employing a hybrid approach that combines time-division multiplexing (TDM) and wavelength-division multiplexing (WDM). This photonic neural network is capable of processing multiple tasks in parallel. As an example, the proposed network includes four layers. Matrix multiplication between the input layer and hidden layer 1 is performed using the TDM method. Subsequent matrix multiplications—from hidden layer 1 to hidden layer 2, and from hidden layer 2 to the output layer—are carried out using the WDM method. A comb source generates multiple wavelengths to facilitate these operations. Complementary metal-oxide-semiconductor (CMOS) comparators are utilized to implement activation functions. TFLN: thin-film lithium niobate.

Line 11, Paragraph 3, Page 11, main manuscript:

“To show the scalability of our solution, we propose an end-to-end photonic neural network that combines the benefits of TDM and WDM methods, as illustrated in Fig. 6. This network is capable of executing multiple AI tasks simultaneously, spanning from the input to the output layer, with nanoseconds latency, all without relying on a digital processor for assistance. As an example, shown in Fig. 6, is a proposed neural network that includes 4 layers: an input layer, two hidden layers, and an output layer.

1) From the input layer to the hidden layer 1. The information of K AI tasks is encoded by K input TFLN modulators and transmitted on K corresponding wavelengths. These signals from input TFLN modulators are then split and channeled into m weighted TFLN modulators. Although some waveguides must pass through $(K - 1)$ crossings, the total insertion losses remain manageable—as an insertion loss of 0.02 dB per crossing has been demonstrated [48]. Following this, each weighted TFLN modulator feeds into a K -channel WDM, which separates the wavelengths to K charge-integration photoreceivers for generating vector-vector dot products. K commercial complementary metal-oxide-semiconductor (CMOS) switches [49] control the output timings of these photoreceivers. Additionally, a CMOS comparator, which selects the maximum between the input and reference voltages, facilitates the ReLU activation function of each vector-vector dot product [50]. The use of 90 wavelengths from a comb source for photonic neural networks [29] and a 64-channel integrated WDM [51] have been previously demonstrated,

making K , $m = 64$ a practical choice. With this setup, we can achieve a computational speed of 491 TOPS and an energy efficiency of 6.5 fJ/OP (i.e., 153 TOPS/W), factoring in a modulation speed of 60 Gbaud/s, including the energy consumption of the laser, DACs, charge-integration photoreceivers, CMOS switches, and CMOS comparators. Further details are available in the Supplementary note 6.

2) From hidden layer 1 to 2, and from hidden layer 2 to the output layer, conventional WDM-based architectures [3, 15] are employed. These parts are unaffected by limitations related to fan-in size, thanks to the relatively small numbers of nodes in the hidden layers. The outputs of hidden layer 1 are fully connected to the m neurons in hidden layer 2. Similarly, the h outputs from hidden layer 2 are fully connected to the p neurons of the output layer, resulting in p network outputs.

This hybrid processor enables the sequential processing of 64 images, each with a resolution of 112×112 pixels, within 62.5 ns. Its significant potential extends to various fields, including autonomous vehicles requiring simultaneous image processing from multiple cameras.”

Comment 3: Overall, I feel that the manuscript has demonstrated an interesting result and the manuscript deserves to be accepted for publication. However, as I have pointed out, the manuscript needs substantial modification and elaboration.

Response: Thank you again for your positive comments.

Comment 1: This manuscript presents an integrated dot product engine that is based on an optoelectronic prototype. It comprises two cascaded TFLN modulators, one hybridly assembled laser connected via a photonic wirebond and two flip-chip photodiodes, using an externally connected electronic assembly circuit for integrating the received signal over time and providing the accumulation operation within the neural network layout. This device is used for demonstrating i) MNIST dataset classification, employing also “hardware-in-the-loop” training, and ii) PCA in the MNIST dataset. Although the prototype is really impressive and its use within a “hardware-in-the-loop” training process is Innovative, I cannot recommend its publication at Nature Communications for reasons explained below:

Response: Thank you for your positive feedback on our prototype and for recognizing the innovative approach to our training process. We also acknowledge and appreciate your constructive criticism. After addressing each point you raised, we are confident that our manuscript has improved considerably.

Comment 2a: 1. The most important reason has to do with the bold claims about the speed, compactness and capacity advantages of their device against state-of-the-art photonic neural network deployments. There are several sections in the paper where these advantages are highlighted, including Introduction (“Our processor significantly surpasses current photonic platforms [9–11, 15, 18, 23, 25], as quantified in Fig. 1c.”, saying then that “our processor achieves a computational speed of 65 GOPS per neuron, including concurrent weight updates—a speed hitherto unachieved, enabling fast “hardware-in-the-loop” training”), and Discussion-Conclusion (“where the performance of our device is compared to that of other state-of-the-art photonic devices [9–11, 15, 18, 23, 25]. Our device excels in all these aspects.”). Fig. D) provides also a pictorial representation of the computational speed versus dimension of dot product for so far demonstrated photonic neural network prototypes. However:

Response: We would also like to point out that this work also claims the ability to perform multiplication with two negative numbers.

Comment 2b: a. regarding speed, this is not the highest speed that is reported in photonic neural network deployment. The authors claim in their introductory paragraph that “To date, all IPTC models are confined to either static (non-volatile) weights, like those utilizing phase change materials [15, 21], or slowly updated (volatile) weights through thermal control [18]”, but the work presented by Giamougiannis et al in Giamougiannis G. et al, “Analog nanophotonic computing going practical: silicon photonic deep learning engines for tiled optical matrix multiplication with dynamic precision” *Nanophotonics*, vol. 12, no. 5, 2023, pp. 963-973. <https://doi.org/10.1515/nanoph-2022-0423> and Giamougiannis G. et al, “Neuromorphic silicon photonics with 50 GHz Tiled Matrix Multiplication for Deep Learning applications”, *SPIE Advanced Photonics*, vol. 5, no. 1, 016004, 2023 employs input and weight updates at 50GHz baudrates, i.e. 100 GOPS, for executing the linear part of multiple neural layers in the optical domain,

Response: Thank you for your comments. Due to the limitation of our previous DAC driver, we could not experimentally demonstrate a computational speed beyond 65GOPS. Following the reviewer’s suggestion, we have tested our processor with a higher-speed driver and have now demonstrated that our processor can achieve a computational speed of 120 GOPS. Theoretically, as shown in Fig. 1c of

the revised manuscript, our solution can achieve a computational speed of over 520 GOPS per neuron according to work reported in Ref. [1], with the vector dimension potentially exceeding over one million.

Thank you also for bringing to our attention the work by Giamougiannis G. et al. Following the reviewer's suggestion, we have revised several sentences in the manuscript and discussed their work. The device experimentally demonstrated by Giamougiannis G. et al. achieved input and weight updates at 50 GHz baud rates, i.e., 100 GOPS. However, their processor performs the dot product for the vectors of only two elements (without assistance from a digital processor for accumulation operations). Theoretically, their solution is scalable to vectors with higher dimensions, but it would require $2N$ modulators for a vector dimension of N .

Our manuscript emphasizes that the reported computational speed is for a *fully integrated* demonstration of a photonic neural network, including laser, detectors, TFLN modulators, and electronics. We believe this is among the first such demonstrations, complementing recent work by Zaijun Chen et al. [2] that just appeared on arXiv.

Furthermore, the discussion section notes that the proposed TDM approach can be combined with other multiplexing schemes (such as WDM), using currently available foundry-compatible technology, to achieve computational speeds of nearly 500 TOPS.

[1] Mardoyan, H., Almonacil, S., Jorge, F., Pittalà, F., Xu, M., Krueger, B., ... & Renaudier, J. (2022, September). First 260-Gb/s single-carrier coherent transmission over 100 km distance based on novel arbitrary waveform generator and thin-film lithium niobate I/Q modulator. In European Conference and Exhibition on Optical Communication (pp. Th3C-2). Optica Publishing Group.

[2] Ou, S., Sludds, A., Hamerly, R., Zhang, K., Feng, H., Zhong, E., ... & Chen, Z. (2024). Hypermultiplexed Integrated Tensor Optical Processor. arXiv preprint arXiv:2401.18050.

Abstract, main manuscript:

“It can execute 120 billion operations per second (GOPS) per neuron,”

Line 16 to Line 24, Paragraph 1, Page 3, main manuscript:

“To date, most IPTC models have been limited to using either static (non-volatile) weights, like those employing phase change materials [15, 21], or volatile weights based on thermo-optic effects [18], which are slow and power inefficient. These methods are unsuitable for “hardware-in-the-loop” training [22]. Although some IPTC models that utilize two cascaded modulators can achieve rapid weight updates, they still require the assistance of a digital computer for accumulation operations or need $2N$ modulators for performing dot product operations on two N -dimensional vectors [23, 24].”

Line 28, Paragraph 3, Page 3, main manuscript:

“our processor achieves a computational speed of 120 GOPS per neuron,”

Line 15 to line 21, Paragraph 2, Page 7, main manuscript:

“The dimension of each vector was set at 131072, a limit imposed by our high-speed arbitrary waveform generator (AWG). The two vectors were modulated separately by two modulators at a modulation rate of 60 Gbaud, enabling a computational speed of 120 GOPS per neuron.”

Fig. 3, main manuscript:

Fig. 3 Experimental result for dot product operation with our device. a A schematic of the working principle of our device. The light is emitted from a laser and then passes through two cascaded thin-film lithium niobate (TFLN) Mach-Zehnder modulators (MZMs). The TFLN MZMs are driven by a high-speed arbitrary waveform generator (AWG). The light is then received by two PDs in a balanced detection scheme, and the corresponding photogenerated electrons are accumulated in the integrator. Reading the output voltage of the integrator with ADC, we can obtain the dot product result. PD: photodetector. ADC: analog-to-digital converter. IC: integrated circuit. DAC: digital to analog converter. b The results of dot product operation between two 131072-dimensional vectors performed by our device with a computational speed of 120 GOPS per neuron. Compared with the expected dot product results, the error of the measured ones has a standard deviation of 0.03 (6.04 bits).

Fig. 4, main manuscript:

Fig. 4 **Classification results of handwritten digits using our device.** **a** A block diagram of a multilayer perceptron neural network, which consists of an input layer, two hidden layers, and an output layer that provides classification outputs. **b** A schematic of the “hardware-in-the-loop” training scheme, a form of online training, where our IPTC handles forward propagation while the computer manages the nonlinearity function and backpropagation. **c** and **d** Theoretically calculated confusion matrices (purely run by the central processing unit (CPU)) and experimental confusion matrices (run by our IPTC) using the MNIST large-scale database [41]. For hardware-in-the-loop training, 2000 handwritten digits are used for training, and 500 digits are used for testing. Our IPTC achieves classification accuracy comparable to that achieved by the CPU.

Comment 2c: b. regarding compactness, the TFLN phase section length is already 1cm, making it hard to imagine how this could form the most compact solution. long and this claim is completely unjustifiable based on the metrics provided in the manuscript. Authors should calculate the area efficiency of their TFLN-based modulation scheme in GOPS/mm² as a metric and compare this to the respective area efficiencies of alternative modulator technologies that could be employed in photonic neural network experiments (i.e. SiGe EAMs, InP MZMs, Silicon Microring Modulators etc) in order to back-up their claim about compactness.

Response: Thank you for your comments. To clarify, we meant that our fully integrated processor is “compact” when compared to photonic neural networks based on free-space [1] or discrete optics [2]. We agree with the reviewer that the footprint of our chip is larger than that of other processors utilizing SiGe EAMs, InP MZMs, and silicon microring modulators. We also agree that the compute density, measured in GOPS/mm², is a more acceptable metric for comparing different processor technologies.

In this light, it has been demonstrated that TFLN modulators can achieve a throughput of 260 Gbaud [3], indicating a compute density of 260 GOPS/mm² with a footprint of 0.2*10 mm². Furthermore, the architecture shown in Fig. 6 of the revised manuscript can achieve a compute density of 7.2 TOPS/mm², with a 64-channel MUX/DEMUX footprint of 0.52×0.62 mm². Moreover, very small TFLN modulators, with lengths around 4 mm, have been recently demonstrated [4].

However, the footprint of fast modulators is not the primary reason for choosing the TFLN technology. One must also account for the size of high-speed drivers (that drive the modulator), which is the bottleneck in achieving an ultra-high compute density. Our preference for TFLN modulators is also rooted in their capacity to: 1) simultaneously offer a low insertion loss, compatibility with CMOS voltages, and a broad Electro-optic bandwidth; 2) Operate across a wide wavelength range, perform multiplications of two negative numbers, and are compatible with hybrid TDM and WDM architectures, as discussed earlier; and 3) have no modulation loss, unlike silicon modulators, which suffer from variable insertion loss depending on the applied voltage.

Following the reviewer's suggestion, we have included a table summarizing and comparing the different technology platforms for our proposed architecture regarding compute density, insertion loss, speed, and other characteristics.

[1] Lin, X., Rivenson, Y., Yardimci, N. T., Veli, M., Luo, Y., Jarrahi, M., & Ozcan, A. (2018). All-optical machine learning using diffractive deep neural networks. *Science*, 361(6406), 1004-1008.

[2] Xu, X., Tan, M., Corcoran, B., Wu, J., Boes, A., Nguyen, T. G., ... & Moss, D. J. (2021). 11 TOPS photonic convolutional accelerator for optical neural networks. *Nature*, 589(7840), 44-51.

[3] Mardoyan, H., Almonacil, S., Jorge, F., Pittalà, F., Xu, M., Krueger, B., ... & Renaudier, J. (2022, September). First 260-GBd single-carrier coherent transmission over 100 km distance based on novel arbitrary waveform generator and thin-film lithium niobate I/Q modulator. In *European Conference and Exhibition on Optical Communication* (pp. Th3C-2). Optica Publishing Group.

[4] Meng, X., Yuan, C., Cheng, X., Yuan, S., Shang, C., Pan, A., ... & Xia, J. (2023). High Performance Thin-film Lithium Niobate Modulator Applied ITO Composite Electrode with Modulation Efficiency of 1V* cm. *arXiv preprint arXiv:2311.05119*.

[5] Li, G. H., Sekine, R., Nehra, R., Gray, R. M., Ledezma, L., Guo, Q., & Marandi, A. (2023). All-optical ultrafast ReLU function for energy-efficient nanophotonic deep learning. *Nanophotonics*, 12(5), 847-855.

Line 6 to Line 17, Paragraph 2, Page 13, main manuscript:

“Compared to photonic processors based on free-space [53] or discrete [29] optics, our fully integrated processor is compact. However, the footprint of fast modulators is not the primary reason for choosing the TFLN technology. The size of high-speed drivers, often the bottleneck in achieving an ultra-high compute density, must also be considered. Our choice of TFLN modulators is driven by their ability to: 1) simultaneously achieve a low insertion loss, CMOS-compatible voltage, and a broad Electro-optic bandwidth [31]; 2) Operate across a wide wavelength range, perform multiplication of two negative numbers, and for their compatibility with hybrid TDM and WDM architectures, as previously discussed; and 3) exhibit no modulation loss, unlike silicon modulators, which suffer from variable insertion loss depending on the applied voltage.”

Line 1 to Line 6, Paragraph 1, Page 9, supplementary:

“Table 1 compares various modulator technologies—SiGe electro-absorption modulator (EAM), InP Mach-Zehnder modulator (MZM), Si MZM, Si microring modulator (MRM)—for our proposed processor architecture. Specifically, Table 1 compares factors like the compute density and insertion loss, among others. Notably, the photonic processor based on TFLN MZM stands out as the only one capable of multiplying with two negative numbers and is fully compatible with hybrid TDM and WDM architectures.”

Table 1, supplementary:

Table S1 Comparison of different modulator technologies employed in our proposed photonic processor which includes two modulators.

Technology	SiGe EAM [3]	InP MZM [4]	Si MZM [5]	Si MRM [6]	TFLN MZM [7]
Modulation speed (Gbaud/s) ^a	50	100	56	32.5	260
Length (mm)	0.61	4	4	0.02	10
Insertion loss (dB/cm) ^b	57	1.5	3	3	0.2
Compute density (GOPs/mm ²)	810	250	140	16250	260
Modulation loss?	N/A	Yes	Yes	Yes	No
Two negative number multiplication?	No	No	No	No	Yes
TDM&WDM? ^c	No	No	No	No	Yes

a. Modulation speed has been demonstrated;

b. Waveguide propagation loss;

c. Whether this technology can be used to build the architecture shown in Fig. 6 of the main manuscript.

N/A indicates no available data.

Comment 2d: c. regarding capacity, this is definitely not a property of the device prototype but of the memory supported by the commercial DACs employed in the experiment.

Response: Thank you for your comments. We realize that using “capacity” can lead to misconceptions. By “capacity,” we meant the “inherent ability” of our fully integrated processor to perform dot product operations without assistance from a digital processor for accumulation operations. Using a charge-integration photoreceiver, our device can execute dot product operations for large vector dimensions without external aid, as shown in Fig. R1(a). With this approach, the computation speed of our processor is not dependent on that of the digital processor (Fig. R1(b)). To avoid any confusion, we have revised the manuscript and replaced the mention of “capacity” with “inherent ability.”

Fig. R1 (a) The architecture implements the charge-integration photoreceiver to perform accumulation operations. (b) The architecture makes use of a digital processor to perform accumulation operations.

Line 39 to Line 40, Paragraph 5, Page 12, main manuscript:

“Our device can simultaneously achieve high performance across all these aspects. Note that “inherent ability” refers to our processor to perform dot product operations without the assistance of a digital processor for accumulation operations.”

Comment 2e: Based on the above, it is in general not clear what are the advantages of using TFLN modulators instead of alternative integrated modulator technology and what are the main advances that this device brings in the field of photonic neural networks.

Response: To highlight the benefits of using TFLN modulators in photonic neural networks, we have prepared a table summarizing the different technology platforms for photonic neural networks regarding compute density, insertion loss, speed, etc. As detailed in our response to Comment 3 earlier, the advantages of TFLN photonics include: 1) to simultaneously achieve a low insertion loss, a CMOS-compatible voltage, and a broad electro-optic bandwidth; 2) to operate across a wide wavelength range, to perform multiplication of two negative numbers, and for their compatibility with hybrid TDM and WDM architectures, as previously discussed; and 3) having no modulation loss, unlike silicon modulators that have insertion losses that vary with the applied voltage.

Line 6 to Line 17, Paragraph 2, Page 13, main manuscript:

“Compared to photonic processors based on free-space [53] or discrete [29] optics, our fully integrated processor is compact. However, the footprint of fast modulators is not the primary reason for choosing the TFLN technology. The size of high-speed DACs, often the bottleneck in achieving an ultra-high compute density, must also be considered. Our choice of TFLN modulators was driven by their ability to: 1) simultaneously achieve low insertion loss, CMOS-compatible voltage, and a broad electro-optic bandwidth [31]; 2) Operate across a wide wavelength range, perform multiplication of two negative numbers, and for their compatibility with hybrid TDM and WDM architectures, as previously discussed; and 3) exhibit no modulation loss, unlike silicon modulators, which suffer from variable insertion loss depending on the applied voltage.”

Comment 3: 2. Probably the most innovative part in this work has to do with the use of the optoelectronic prototype in the “hardware-in-the-loop” training process. However, more information is needed in order to understand all the details of this demonstration. The authors should provide details about the (1) interfaces employed between the CPU and the optoelectronic assembly, how was the (2) model initialized and how were the normalization stages implemented, how the (3) digital weight update values obtained from the CPU were translated into analog voltages, (4) how many epochs were required until converging creating a graph that shows the convergence over time for both the purely digital and the “hardware-in-the-loop” training process would be helpful, (5) how did the training model deal with the noise of the optoelectronic system, (6) how does this solution compare to the training of the respective model exclusively in the digital domain, what was the bit resolution per neural layer required when training in the digital domain etc. A graph showing how the performance of the training process varies when the bit resolution of every individual layer drops below the ideal value would be very helpful.

Response: Per the reviewer's suggestion, we have provided more details in the revised manuscript.

1) A schematic of interfaces employed between the CPU and the optoelectronic assembly is shown in Fig. S5a of the revised supplementary materials. We use Python, running on the CPU, to control all devices. As illustrated in Fig. S5b of the revised supplementary materials, when a matrix-vector multiplication operation is required, the CPU sends a "Start to Integration" command to the charge integrator. At this point, no charge is accumulated due to the balancing scheme. Subsequently, the CPU sends a "Start to Encode" command to the DACs. Once all data is encoded, the CPU sends an "End to Integration" command to the charge integrator. The integrated circuit (IC) then sequentially returns the voltage from the charge integrator to the CPU, enabling the retrieval of the matrix-vector multiplication result.

2) The DACs perform the digital-to-analog conversion and map the digital input vector and weight updates to their equivalent analog representation. With a V_{π} of 2.4 V, the modulator is biased to achieve a voltage swing between -1.2 V and +1.2 V, allowing for a direct mapping of weights from -1 to +1.

3) The CPU maps the input vector and weight values to the 8-bit DAC: 1 bit for the sign and 7 bits (128 discrete values) for the magnitude of the vector/weight.

4) Fig. S7a of the revised manuscript shows the convergence over time for both purely digital and “hardware-in-the-loop” configurations.

5) One advantage of hardware-in-the-loop training is that it inherently accounts for hardware nonidealities, including fabrication variations and noise in the system. Essentially, the nonidealities are “baked into” the training process. This approach has been experimentally demonstrated in Refs. [1,2].

6) As shown in Fig. S7b of the revised Supplementary, 4 bits of precision can be sufficient for neural network computation, as also demonstrated in Ref. [1].

[1] Filipovich, M. J., Guo, Z., Al-Qadasi, M., Marquez, B. A., Morison, H. D., Sorger, V. J., ... & Shastri, B. J. (2022). Silicon photonic architecture for training deep neural networks with direct feedback alignment. *Optica*, 9(12), 1323-1332.

[2] Buckley, S.M., Tait, A.N., McCaughan, A.N., Shastri, B.J.: Photonic online learning: a perspective.

Line 4 to Line 7, Paragraph 1, Page 8, main manuscript:

“Compared with the expected dot product result, the error of the measured one has a standard deviation of 0.03 (6.04 bits)—more than the 4 bits of precision required for performing AI tasks (details can be found in Supplementary note [40]).”

Line 17 to Line 22, Paragraph 3, Page 9, main manuscript:

“Our IPTC achieved near theoretical accuracy, indicating that the “hardware-in-the-loop” training scheme enables the system to inherently account for the hardware nonidealities, including fabrication variations and noise. Essentially, the nonidealities are “baked into” the training process. This has also been experimentally demonstrated in Ref. [3].”

Line 11 to Line 13, Paragraph 2, Page 9, main manuscript:

*“More details regarding the training algorithm, the interfaces between the central processing unit (CPU) and the optoelectronic assembly, and convergence speed can be found in the **Methods** and **Supplementary note 4**.”*

Line 12 to Line 20, Paragraph 2, Page 5, Supplementary:

“We use Python, running on the CPU, to control all devices. As illustrated in Fig. S5b of the revised supplementary materials, when a dot product operation is required, the CPU sends a “Start to Integration” command to the charge integrator. However, at this point, no charge is being accumulated due to the balancing scheme. Subsequently, the CPU sends a “Start to Encode” command to the DACs. Once all data is encoded, the CPU sends an “End to Integration” command to the charge integrator. The integrated circuit (IC) then returns the voltage of the charge integrator to the CPU sequentially, enabling the retrieval of the dot product result.”

Line 1 to Line 7, Paragraph 1, Page 7, Supplementary:

“The test accuracy as a function of the effective resolution used in the matrix-vector multiplication operations is shown in Fig. S6a. These simulation results demonstrate that the training process is robust to noise and low-precision computations. Therefore, although our processor only has a resolution of 6.04 bits, it is sufficient for training. The validation accuracy during training, using both our processor and a CPU (for comparison), is depicted in Fig. S6b. This demonstrates that the “hardware-in-the-loop” training approach can achieve a convergence speed comparable to that of training solely on the CPU.”

Fig. S5, Supplementary:

Fig. S5 Experimental setup for characterizing the performance of our device. A schematic of the experimental setup used for dot product demonstration, classification, and clustering. AWG: arbitrary waveform generator.

Fig. S6, Supplementary:

Fig. S6 **a** Validation accuracy on the MNIST dataset as a function of the effective resolution. **b** The convergence over epoch for both “hardware-in-the-loop” training and pure central processing unit (CPU).

Comment 4: 3. Authors should provide more details on how the energy efficiency of 800 fJ per operation has been calculated and what this includes. Are DACs and drivers taken into account, as this should be the case in order to allow for a fair comparison with alternative neural network hardware deployment? A detailed breakdown of the energy consumption should be provided. Moreover, one of the main motivations for photonic neural network developments is their potential to drive energy efficiency to values lower than current state-of-the-art, well below sub-pJ/MAC. Can the proposed scheme scale towards supporting competitive energy efficiency values and, if yes, how?

Response: Thank you for your questions. In theory, the energy efficiency of 828 fJ/OP includes the laser driver, DAC, bias voltage controller, op-amp integrator, and ADC. The laser driver, DAC, op-amp integrator, and ADC specifications are sourced from Refs. [1-4], respectively. By improving the structure of heaters used for bias control and increasing the modulation rate up to 60 Gbaud, the

proposed photonic tensor core can achieve an energy efficiency of 213 fJ/OP. Moreover, as shown in Table 2, the proposed scalable photonic tensor core, which includes 64 + 64 TFLN modulators, can achieve an energy efficiency of 6.5 fJ/OP with a computational speed of 526.5 TOPS.

- [1] Sackinger, E., Ota, Y., Gabara, T. J., & Fischer, W. C. (2000). A 15-mW, 155-Mb/s CMOS burst-mode laser driver with automatic power control and end-of-life detection. *IEEE Journal of Solid-State Circuits*, 35(2), 269-275.
- [2] Nguyen, R. L., Castrillon, A. M., Fan, A., Mellati, A., Reyes, B. T., Abidin, C., ... & Elsharkasy, W. (2021, February). A Highly Reconfigurable 40-97GS/s DAC and ADC with 40GHz AFE Bandwidth and Sub-35fJ/conv-step for 400Gb/s Coherent Optical Applications in 7nm FinFET. In *2021 IEEE International Solid-State Circuits Conference (ISSCC)* (Vol. 64, pp. 136-138). IEEE.
- [3] Yang, E., & Lehmann, T. (2019, May). High gain operational amplifiers in 22 nm CMOS. In *2019 IEEE International Symposium on Circuits and Systems (ISCAS)* (pp. 1-5). IEEE.
- [4] <https://github.com/bmurmman/ADC-survey>

Line 1 to Line 5, Paragraph 1, Page 10, Supplementary:

“In theory, for a computation speed of 120 GOPS, our device can achieve an energy efficiency of 213 fJ/OP (see Table 2), including the laser driver, DAC, bias voltage controller, op-amp integrator, and ADC. Moreover, as shown in Table 3, the proposed scalable photonic tensor core, which includes 64 + 64 TFLN modulators, can achieve an energy efficiency of 6.5 fJ/OP with a computational speed of 491 TOPS (i.e., 153 TOPS/W). We also compare our proposed solution with some other optical processors in Table S4.”

Table 2, Supplementary:

Table S2 Energy of our proposed photonic processor which includes two TFLN modulators.

Components	Energy budget	Energy efficiency (32.5 Gbaud)	Energy efficiency (60 Gbaud)
Laser driver [8]	15 mW	230 fJ/OP	125 fJ/OP
DAC (7nm FinFET) [9]	35 fJ/conv-step	35 fJ/OP	29 fJ/OP
Bias controller	17 mW	523 fJ/OP	56 fJ/OP ^a
Integrator [10]	44 μ W	0.67 fJ/OP	0.37 fJ/OP
ADC (1 GHz) [10]	2.55 mW	39 fJ/OP	2.17 fJ/OP
Total	N/A	828 fJ/OP	213 fJ/OP

a. Heating efficiency can be improved up to 3 times higher than the presented structure [11].

N/A indicates no available data

Table 3, Supplementary:

Table S3 Energy of our proposed photonic processor which includes 64 + 64 TFLN modulators with a computation speed of 491 TOPS.

Components	Energy budget	Number	Energy efficiency (60 Gbaud)
Pump laser	1 W	1	2.04 fJ/OP
DAC (7nm FinFET) [9]	35 fJ/conv-step	64 + 64	0.59 fJ/OP
Bias controller	5.6 mW	64 + 64	1.46 fJ/OP ^a
Integrator [10]	44 μ W	64 \times 64	0.37 fJ/OP
CMOS comparator [12]	244.19 μ W	64 \times 64	2.04 fJ/OP
CMOS switch [13]	122 nW	64 \times 64	0.001 fJ/OP
Total	N/A	N/A	6.5 fJ/OP

a. Heating efficiency can be improved up to 3 times higher than the presented structure [11].

N/A indicates no available data.

Table 4, Supplementary:

Table S4 Comparison with some optical computing architectures both on-chip and in free space. N/A indicates no available data.

Sources	Energy efficiency ^a	Total neuron ^a	Tunable neuron	Network scale
This work ^b	153 TOPS/W	802,816	802,816	51 million
Feldmann et al. [14]	0.50 TOPS/W	64	64	29,186
Ashtiani et al. [15]	2.9 TOPS/W	67	67	67
Shen et al. [16]	N/A	213	213	1065
Zhou et al. [17]	0.71 TOPS/W	490,000	490,000	1.7 million
Xu et al. [18]	N/A	N/A	867	867
NVIDIA H100 PCIe [19]	0.71 TOPS/W	N/A	N/A	N/A

a. By theoretical calculations.

b. Based on our proposed photonic processor which includes 64 + 64 TFLN modulators.

Comment 5: 4. The Introduction section refers at the end of its second paragraph: "...Additionally, IPTCs based on interferometric meshes [8, 18] require a single laser source but face scalability issues due to the multitude of directional couplers and phase shifters involved. To date, all IPTC models are confined to either static (non-volatile) weights, like those utilizing phase change materials [15, 21], or slowly updated (volatile) weights through thermal control [18], which are power intensive, rendering them unsuitable for "hardware-in-the-loop" training [22]. While these platforms can achieve high computational speeds, they are limited to small-scale vector operations and inference tasks." The state-of-the-art on interferometric meshes should be enriched to include developments and architectures demonstrated by the groups of O. Liboiron-Ladouceur, N. Pleros and N. Youngblood. Moreover, additional important work regarding the TDM scheme should be referenced, like the work of L. DeMarinis et al., "A Codesigned Integrated Photonic Electronic Neur"n," in IEEE Journal of Quantum Electronics, vol. 58, no. 5, pp. 1-10, Oct. 2022, Art no. 8100210, doi: 10.1109/JQE.2022.3177793 and

Z. Zhong et al, “Lightning: A Reconfigurable Photonic-Electronic SmartNIC for Fast and Energy-Efficient Inference”, in Proc. of the ACM SIGCOMM 2023 Conference (ACM SIGCOMM '23) New York, NY, USA, 452–472, <https://doi.org/10.1145/3603269.3604821> .

Response: Based on the reviewer’s suggestion, we have rewritten these sentences and cited the references mentioned by the reviewer in the revised manuscript.

Line 16 to Line 24, Paragraph 1, Page 3, main manuscript:

“To date, most IPTC models have been limited to using either static (non-volatile) weights, like those employing phase change materials [15, 21], or volatile weights based on thermo-optic effects [18], which are slow and power inefficient. These methods are unsuitable for “hardware-in-the-loop” training [22]. Although some IPTC models that utilize two cascaded modulators can achieve rapid weight updates, they still require the assistance of a digital computer for accumulation operations or need $2N$ modulators for performing dot product operations on two N -dimensional vectors [23, 24].”

Line 25 to Line 26, Paragraph 2, Page 3, main manuscript:

“Recently, photocurrent integrators have been proposed for accumulation operations in a time-division multiplexing (TDM) scheme [25–27].”

Comment 6: 5. The Discussion section presents a conceptual layout of a possible architecture that combines TDM and WDM. It is a very high-level setup that exploits techniques that have not been validated so far as such and, for that reason, it is rather questionable on whether this comprises indeed a valid perspective. For example, the weighting stage at the Input Layer exploits TFLN modulators for weighting multiple input channels simultaneously, which has not been demonstrated experimentally and raises several concerns regarding its feasibility and performance. Is it possible in this case, for example, to apply again the concept for two negative number multiplication? On the same line, the rather simplistic assumption of utilizing 90 wavelengths from a single comb source towards reaching 500TOPs builds upon i) a broadband operational regime for the TFLN modulators, ii) a high amount of 90 cascaded resonant devices in hidden layer #2 and in the output layer, and iii) 90-ch MUX/DEMUX elements. Moreover, there are several waveguide crossings and delays foreseen in this layout, which are difficult to envision in integrated form. The total losses of the layout and the noise are also not discussed at all. This section should be rewritten so as to extend along a more scientifically and technologically validated roadmap, including either simulation results or at least details about every of its stages and components, or alternatively should be omitted in order to avoid misleading conclusions.

Response: Thank you for your comments.

1) Multiplication of two negative numbers can be achieved in large-scale architectures due to the broadband operational regime of the TFLN modulators [1]. Including this feature in the proposed scalable architecture shown in Fig. 6 of the revised manuscript would significantly increase its complexity. This addition could obscure the main message, which is to show scalability through hybrid TDM and WDM schemes.

2) In this manuscript, we have proposed utilizing 90 wavelengths from a single comb source for our photonic tensor core, as the effectiveness of using 90 wavelengths from a comb source has already been demonstrated for optical neural networks in Ref. [1]. Additionally, 90-channel multiplexer/

demultiplexer (MUX/DEMUX) elements are commercially available [3].

3) To avoid misunderstanding, we have replaced “delay line” with “CMOS switch” in Fig. 6 of the revised manuscript.

4) We have rewritten this section following the reviewer’s suggestion.

[1] Ke, W., Lin, Y., He, M., Xu, M., Zhang, J., Lin, Z., ... & Cai, X. (2022). Digitally tunable optical delay line based on thin-film lithium niobate featuring high switching speed and low optical loss. *Photonics Research*, 10(11), 2575-2583.

[2] Xu, X., Tan, M., Corcoran, B., Wu, J., Boes, A., Nguyen, T. G., ... & Moss, D. J. (2021). 11 TOPS photonic convolutional accelerator for optical neural networks. *Nature*, 589(7840), 44-51.

[3] <https://agiltron.com/product/96-channels-ft/>.

Fig. 6, main manuscript:

Fig. 6 A schematic of a photonic neural network designed to show the scalability of the proposed integrated photonic tensor core, employing a hybrid approach that combines time-division multiplexing (TDM) and wavelength-division multiplexing (WDM). This photonic neural network is capable of processing multiple tasks in parallel. As an example, the proposed network includes four layers. Matrix multiplication between the input layer and hidden layer 1 is performed using the TDM method. Subsequent matrix multiplications—from hidden layer 1 to hidden layer 2, and from hidden layer 2 to the output layer—are carried out using the WDM method. A comb source generates multiple wavelengths to facilitate these operations. Complementary metal-oxide-semiconductor (CMOS) comparators are utilized to implement activation functions. TFLN: thin-film lithium niobate.

Line 11, Paragraph 3, Page 11, main manuscript:

“To show the scalability of our solution, we propose an end-to-end photonic neural network that combines the benefits of TDM and WDM methods, as illustrated in Fig. 6. This network is capable of executing multiple AI tasks simultaneously, spanning from the input to the output layer, with

nanoseconds latency, all without relying on a digital processor for assistance. As an example, shown in Fig. 6, is a proposed neural network that includes 4 layers: an input layer, two hidden layers, and an output layer.

1) From the input layer to the hidden layer 1. The information of K AI tasks is encoded by K input TFLN modulators and transmitted on K corresponding wavelengths. These signals from input TFLN modulators are then split and channeled into m weighted TFLN modulators. Although some waveguides must pass through $(K-1)$ crossings, the total insertion losses remain manageable, as an insertion loss of 0.02 dB per crossing has been demonstrated [48]. Following this, each weighted TFLN modulator feeds into a K -channel WDM, which separates the wavelengths to K charge-integration photoreceivers for generating vector-vector dot products. K commercial complementary metal–oxide–semiconductor (CMOS) switches [49] control the output timings of these photoreceivers. Additionally, a CMOS comparator, which selects the maximum between the input and reference voltages, facilitates the ReLU activation function of each vector-vector dot product [50]. The use of 90 wavelengths from a comb source for photonic neural networks [29] and a 64-channel integrated WDM [51] have been previously demonstrated, making $K, m = 64$ a practical choice. With this setup, we can achieve a computational speed of 491 TOPS and an energy efficiency of 6.5 fJ/OP (i.e., 153 TOPS/W), factoring in a modulation speed of 60 Gbaud/s, including the energy consumption of the laser, DACs, charge-integration photoreceivers, CMOS switches, and CMOS comparators. Further details are available in the Supplementary note 6.

2) From hidden layer 1 to 2, and from hidden layer 2 to the output layer, conventional WDM-based architectures [3, 15] are employed. These parts are unaffected by limitations related to fan-in size, thanks to the relatively small numbers of nodes in the hidden layers. The outputs of hidden layer 1 are fully connected to the m neurons in hidden layer 2. Similarly, the h outputs from hidden layer 2 are fully connected to the p neurons of the output layer, resulting in p network outputs.

This hybrid processor enables the sequential processing of 64 images, each with a resolution of 112×112 pixels, within 62.5 ns. Its significant potential extends to various fields, including autonomous vehicles requiring simultaneous image processing from multiple cameras.”

Sincerely yours,
Lukas Chrostowski

REVIEWER COMMENTS

Reviewer #1 (Remarks to the Author):

I have no further comments on the paper. However, I feel that reviewer 2's comments are absolutely valid and the authors should address them thoroughly. Lithium Niobate is not the cheapest material for large scale production but as a proof of principle I think the paper has made a step forward.

Reviewer #2 (Remarks to the Author):

The authors have invested considerable effort and have improved their manuscript. They even repeated the experiments to demonstrate 120 GOPS/neuron instead of the 65 GOPs they had before, which is highly appreciated. However, I think that the novelty of this work is rather marginal and after the revision process mainly refers to:

1. The experimental deployment and demonstration of a concept that is not new and was originally proposed by L. De Marinis et al (ref 26 in the manuscript). The layout of De Marinis et al in [24] is identical to the circuit design demonstrated by the authors, supporting also the use of negative numbers and the accumulation via analog electronics without the use of digital electronics. Moreover, although the authors have referenced this work in their revised manuscript, they should certainly mention explicitly that their work relies on the concept and layout that was originally proposed in [24].
2. The highest operational rate per neuron (120GOPS), improving by 20% the 100 GOPS presented by G. Giamougiannis et al in refs 23 and 24 in the manuscript). However, this 20% improvement in baudrate (from the 50 Gbaud to 60Gbaud) forms again a rather marginal improvement, when taking also into account the current capabilities of integrated modulator technologies. In addition, the authors claim 120GOPS/neuron as being the highest rate per neuron, which is not true. Their rate is indeed the highest rate per channel or per axon, but a neuron comprises several converged axons; as such, the work of G. Giamougiannis et al in refs 23 and 24 has a rate per neuron that equals 200GOPS, given that this work includes two branches/axons with every axon performing at 100GOPS and both axons summing together to form a neuron. On top of that, a new record computational rate per neuron equal to 640 GOPS was demonstrated in the meantime as an OFC 2024 Post-deadline paper, see C. Pappas et al, "A 160 TOPS Multi-dimensional AWGR-based accelerator for Deep Learning".
3. The experimental demonstration of the "hardware-in-the-loop" approach. This is the most interesting and novel part of this work, but the manuscript gives the impression that its focus relies rather on items #1 and #2 above without emphasizing on the "hardware-in-the-loop" demonstration. As this is currently described in the manuscript, this cannot be considered enough for a publication at Nat Comms.

The overall feeling is that this work is a remarkable engineering achievement but is rather poor with respect to its scientific novelty and, to my opinion, not at the level required for a Nature Communications article. For that reason, I don't think that this can be accepted for publication at Nat Comms and I would encourage the authors to submit to a different journal that has a stronger focus on engineering, like JLT, JSTQE, JOCN etc.

Type: Research article

Title: 120 GOPS/neuron Photonic Tensor Core in Thin-film Lithium Niobate for Inference and In-situ-training

Dear Reviewers,

We want to thank you for your time in reviewing our manuscript (NCOMMS-23-57567A). We found the reviewers' report very insightful. In response, we have carefully revised our paper in accordance with the comments provided by the editor and reviewers. Enclosed, please find our detailed point-by-point response to each comment. We believe that our manuscript has been significantly strengthened as a result.

Specifically, to improve the quality and impact of our revised manuscript, we have clarified the novelty of our work based on the reviewers' suggestions. Below is our point-by-point response to the referees' reports. For clarity, the reviewers' comments are presented in **blue text**, our responses in black, and quotations from our revisions (including all figures from our revisions) in *black italic text*

Reviewer #1

Comment: I have no further comments on the paper. However, I feel that reviewer 2's comments are absolutely valid and the authors should address them thoroughly. Lithium Niobate is not the cheapest material for large scale production but as a proof of principle I think the paper has made a step forward.

Response: Thank you for your positive comments. We have addressed all of reviewer 2's comments and hope that you find our manuscript suitable for publication.

Reviewer #2

Comment 1: The authors have invested considerable effort and have improved their manuscript. They even repeated the experiments to demonstrate 120 GOPS/neuron instead of the 65 GOPs they had before, which is highly appreciated. However, I think that the novelty of this work is rather marginal and after the revision process mainly refers to:

Response: Increasing the speed of our platform from 65 GOPS to 120 GOPS during the last revision was a substantial effort. Thank you for appreciating it. To address the reviewer's comment, we would like to further clarify the novelty of our work, which includes:

(1) The demonstration of the first *fully integrated* photonic tensor core with lasers, photodetectors, thin-film lithium niobate (TFLN) chips, and electronics. One of the main challenges in TFLN photonics is fabricating the lasers and photodetectors. Here, we present a solution for full integration.

(2) Our device is the first to provide a solution for multiplications involving two negative numbers. This capability enables in-situ training for image clustering tasks based on principal component analysis. Note that other solutions can only perform multiplications involving a positive number and a negative number or two positive numbers. (*Hardware-in-the-loop training* can also be referred to as in-situ training. In the revised manuscript and the following text, we will use the term in-situ training.)

(3) Our experimental demonstration of in-situ training entails a weight update speed of 60 GHz and a computational speed of 120 GOPS for dot-product operations with vector dimensions up to 131,072, without the assistance of digital computing or any postprocessing.

(4) We propose a solution that combines the benefits of TDM and WDM methods, allowing the simultaneous execution of multiple end-to-end AI tasks with nanosecond latency without relying on a

digital processor. With scaling in space and wavelength, similar to the architecture described in the conference paper by C. Pappas et al. (2024) and as proposed in our manuscript, our system has the potential to scale to 500 TOPS with an end-to-end photonic neural network.

(5) We experimentally demonstrate the versatility of our photonic tensor core in performing a wide range of AI tasks. These tasks include classification, which is part of supervised learning, and clustering, which is part of unsupervised learning. Our photonic tensor core handles these diverse tasks successfully, highlighting its flexibility and suitability for various AI applications.

Comment 2: 1. The experimental deployment and demonstration of a concept that is not new and was originally proposed by L. De Marinis et al (ref 26 in the manuscript). The layout of De Marinis et al in [24] is identical to the circuit design demonstrated by the authors, supporting also the use of negative numbers and the accumulation via analog electronics without the use of digital electronics. Moreover, although the authors have referenced this work in their revised manuscript, they should certainly mention explicitly that their work relies on the concept and layout that was originally proposed in [24].

Response: Thank you for your suggestion. First, we would like to emphasize that the work by De Marinis et al. in [24] is based on a simulation. We have experimentally demonstrated our system at 120 GOPS and furthermore fully integrated our processor with lasers, detectors, and cascaded modulators, including the electronics. Second, the layout presented in [24] can only perform multiplication involving a positive number and a negative number, or two positive numbers. As shown in Fig. 5a of the revised manuscript, our layout, to the best of our knowledge, is the first that can perform multiplications involving two negative numbers. To clarify these points, we have rewritten the corresponding sentences in the revised manuscript to show that the idea of performing accumulation via analog electronics was originally proposed in [24].

Line 1 to Line 3, Paragraph 2, Page 3, main manuscript:

“Recently, a solution first proposed by L. De Marinis et al. [25], which uses photocurrent integrators to perform accumulation operations in a time-division multiplexing (TDM) scheme, has been gaining increasing attention [26, 27].”

Line 3 to Line 5, Paragraph 3, Page 3, main manuscript:

“Notably, to the best knowledge, our device is the first to provide a solution for multiplications involving two negative numbers, thanks to the ability of TFLN modulators to operate across a wide wavelength range.”

Fig. 5, main manuscript:

Fig. 5 Clustering result of the handwritten digits using our device. *a* A schematic of the working principle of our device to perform the multiplication between two numbers with any signs, including two negative numbers. MZM: Mach-Zehnder modulator. *b* The variance of the projected points, X_{bi} , as a function of iterations when finding the first principle component using the power method. X is a $p \times n$ data matrix, representing the 10000 of the handwritten digits from the MNIST large-scale database [40]. b_i is a $n \times 1$ unit vector, obtained at the i th iteration. The algorithm performed by our device has a comparable iteration speed with that of the central processing unit (CPU). *c* and *d* are the front and rear views of the 3D coordinates of each handwritten digit based on the scores of projecting onto the first three principal components (PCs), respectively.

Comment 3: 2. The highest operational rate per neuron (120GOPS), improving by 20% the 100 GOPS presented by G. Giamougiannis et al in refs 23 and 24 in the manuscript). However, this 20% improvement in baudrate (from the 50 Gbaud to 60 Gbaud) forms again a rather marginal improvement, when taking also into account the current capabilities of integrated modulator technologies. In addition, the authors claim 120 GOPS/neuron as being the highest rate per neuron, which is not true. Their rate is indeed the highest rate per channel or per axon, but a neuron comprises several converged axons; as such, the work of G. Giamougiannis et al in refs 23 and 24 has a rate per neuron that equals 200 GOPS, given that this work includes two branches/axons with every axon performing at 100 GOPS and both axons summing together to form a neuron. On top of that, a new record computational rate per neuron equal to 640 GOPS was demonstrated in the meantime as an OFC 2024 Post-deadline paper, see C. Pappas et al, "A 160 TOPS Multi-dimensional AWGR-based accelerator for Deep Learning".

Response: Thank you for your comment. We realize that different terminology (“axon,” “synapse,” and “channel”) used by various groups can create confusion and complicate direct comparisons. Therefore, we have removed the claim regarding “the highest rate per neuron” in the revised manuscript. The key points of comparison are the speed for each modulator, the currently demonstrated vector scale, and the current level of integration.

The three architectures (ours and the two mentioned) all propose paths to scale to higher numbers of OPS, but ours demonstrates the fastest performance per channel. We are 20% faster per modulator than Giamougiannis et al.’s 200 GOPS claim, but their architecture suffers from poor scaling due to the requirement of $2N$ modulators for performing dot product operations on two N -dimensional vectors if they avoid the assistance of a digital processor. Additionally, relying on a digital processor introduces latency, limiting the overall computational speed. Our modulators are 3x faster per modulator than the

640 GOPS claim by C. Pappas et al. Furthermore, their two architectures require summing in the digital domain, which limits their effective speed and is not mentioned in their manuscripts. In contrast, our architecture surpasses both approaches in terms of modulation speed, progress toward integration, summing at full speed in the analog electrical domain, and scalability, as shown in Fig. 6.

In addition to computational speed, other characteristics of a photonic tensor core are also important, such as the fan-in size for matrix-vector multiplications, fast weight update speed for in-situ training, and more. Please see the novelty claims (full integration, in-situ training, etc.) as detailed in our response to comment 1 above.

Reference, main manuscript:

“[25] Pappas, C., Moschos, T., Prapas, A., Tsakyridis, A., Moralis-Pegios, M., Vyrsoinos, K., Pleros, N.: A 160 TOPS multi-dimensional AWGR based accelerator for deep learning. In: *Optical Fiber Communication Conference*, pp. 4–3 (2024). Optica Publishing Group.

[26] Pappas, C., Moschos, T., Moralis-Pegios, M., Giamougiannis, G., Tsakyridis, A., Kirtas, M., Passalis, N., Tefas, A., Pleros, N.: A TeraFLOP photonic matrix multiplier using time-space-wavelength multiplexed AWGR-based architectures. In: *2024 Optical Fiber Communications Conference and Exhibition (OFC)*, pp. 1–3 (2024). IEEE.”

Comment 4: 3. The experimental demonstration of the "in-situ" approach. This is the most interesting and novel part of this work, but the manuscript gives the impression that its focus relies rather on items #1 and #2 above without emphasizing on the "hardware-in-loop" demonstration. As this is currently described in the manuscript, this cannot be considered enough for a publication at Nat Comms.

Response: Thank you for agreeing that the in-situ approach is one of the novel parts of our work. In response to the reviewer’s suggestion, we have added more content to emphasize the hardware-in-loop training demonstration in the revised manuscript. We have changed the title, rewritten portions of the abstract, added Fig. 4c, and made other modifications.

Title:

“120 GOPS/neuron Photonic Tensor Core in Thin-film Lithium Niobate for Inference and In-situ training”

Abstract:

“Our processor supports fast in-situ training with a weight update speed of 60 GHz.”

Abstract:

“Furthermore, it successfully classifies (supervised learning) and clusters (unsupervised learning) 112×112-pixel images through in-situ training. To facilitate in-situ training for clustering AI tasks, we provide a solution for multiplications involving two negative numbers enabled by our processor.”

Line 1 to Line 7, Paragraph 3, Page 2, main manuscript:

“Rapid weight updates: Accurate and efficient training necessitates the use of in-situ training (i.e., in-situ training) [1–4]. This method incorporates real-time feedback from processors into the weight update loop, accounting for processor imperfections and environmental changes. Rapid weight updates speed up training and facilitate “on-the-fly” or online learning, which is particularly beneficial for applications such as autonomous vehicles [5].”

Line 15 to Line 23, Paragraph 1, Page 3, main manuscript:

“To date, most IPTCs have been limited to using either static (non-volatile) weights, such as those employing phase change materials [15, 21], or volatile weights based on thermo-optic effects [18], which are slow and power inefficient. These methods are unsuitable for “in-situ” training [22].”

Although some IPTC models that utilize two cascaded modulators can achieve rapid weight updates, they still require summing in the digital domain which strongly limits the final computational speed, or they need $2N$ modulators for performing dot product operations on two N -dimensional vectors [23, 24].”

Line 3 to Line 5, Paragraph 3, Page 3, main manuscript:

“Moreover, with a weight update speed of 60 GHz, our processor enables fast in-situ training.”

Line 18 to Line 20, Paragraph 2, Page 4, main manuscript:

“Moreover, through this architecture, our device can perform fast in-situ training because it can update the weight vectors at the modulation speed of the modulator.”

Line 7 to Line 10, Paragraph 2, Page 7, main manuscript:

“The two vectors were modulated separately by two modulators at a modulation rate of 60 Gbaud, enabling a computational speed of 120 GOPS per neuron, and a weight update speed of 60 GHz.”

Line 9 to Line 10, Paragraph 2, Page 9, main manuscript:

“Fig. 4c shows the validation accuracy as a function of epoch for in-situ training scheme compared to that running on just a central processing unit (CPU).”

Fig. 4, main manuscript:

Fig. 4 Classification results of handwritten digits using our device. **a** A block diagram of a multilayer perceptron neural network, which consists of an input layer, two hidden layers, and an output layer that provides classification outputs. **b** A schematic of the “in-situ” training, a form of online training, where our IPTC handles forward propagation while the computer manages the nonlinearity function and backpropagation. **c** The validation accuracy as a function of epoch for in-situ training (solid red line) scheme compared to that running on just a central processing unit (CPU, dashed blue line). **d** and **e** Theoretically calculated confusion matrices (purely run by the CPU) and experimental confusion matrices (run by our IPTC) using the MNIST large-scale database [40]. For “in-situ” training, 2000 handwritten digits are used for training, and 500 digits are used for testing. Our IPTC achieves classification accuracy comparable to that achieved by the CPU.

Comment 5: The overall feeling is that this work is a remarkable engineering achievement but is rather poor with respect to its scientific novelty and, to my opinion, not at the level required for a Nature Communications article. For that reason, I don't think that this can be accepted for publication at Nat Comms and I would encourage the authors to submit to a different journal that has a stronger focus on engineering, like JLT, JSTQE, JOCN etc.

Response: As discussed through our responses to all the comments, we believe our manuscript meets the quality standards required for publication in *Nature Communications*. Although De Marinis et al. [24] proposed the idea of using analog electronics to perform accumulation operations, they only demonstrated this through simulations. Transitioning from simulations to an experimental demonstration is a significant challenge, which requires addressing numerous scientific and technological problems.

In addition to the experimental demonstration of analog electronics-based accumulation operation, our work also provides a fully integrated demonstration of a photonic tensor core. It shows the capability of performing multiplications involving two negative numbers, and experimentally demonstrates that our device can process both supervised and unsupervised learning AI tasks through in-situ training, among other claims (please see response to comment 1).

Using analog electronics to perform accumulation operations is only part of our work. Therefore, although De Marinis et al. have proposed this idea, we respectfully state that this does not negate the scientific novelty of our work nor justify its rejection by *Nature Communications*. For example, G. Giamougiannis et al. proposed a photonic neural network architecture in references 23 and 24, yet they published a work with a similar architecture in *Nature Communications* (15.1 (2024): 5468) last month.

We trust that the revisions we have implemented address the reviewers' concerns and effectively underscore the unique contributions of our research. We look forward to your favorable consideration for publication.

Sincerely yours,

Lukas Chrostowski (on behalf of the co-authors)

REVIEWERS' COMMENTS

Reviewer #2 (Remarks to the Author):

The authors have significantly improved their manuscript and have provided significantly more details regarding the in-situ training, which is also the most innovative part of their work. As such, I believe that their work can be now published, provided, however, that they change the term 120GOPS/neuron in their title, which is misleading.

The operation on a per neuron level is completed only once the summation is also completed, which in their case doesn't happen at 120GHz since their scheme follows the TDM approach. They should instead use the term 120GOPS without the "per neuron" indication in the title, since this is their computational line-rate (line-rate of multiplications) but not the operational rate per neuron.

Type: Research article

Title: 120 GOPS Photonic Tensor Core in Thin-film Lithium Niobate for Inference and in-situ Training

Dear Reviewers,

We want to thank you for your time in reviewing our manuscript (NCOMMS-23-57567B). We found the reviewers' report very insightful. In response, we have carefully revised our paper in accordance with the comments provided by the editor and reviewers. Enclosed, please find our detailed point-by-point response to each comment. We believe that our manuscript has been significantly strengthened as a result.

Specifically, to improve the quality and impact of our revised manuscript, we have clarified the novelty of our work based on the reviewers' suggestions. Below is our point-by-point response to the referees' reports. For clarity, the reviewers' comments are presented in **blue text**, our responses in black, and quotations from our revisions (including all figures from our revisions) in *black italic text*

Reviewer #2

Comment 1: The authors have significantly improved their manuscript and have provided significantly more details regarding the in-situ training, which is also the most innovative part of their work. As such, I believe that their work can be now published, provided, however, that they change the term 120GOPS/neuron in their title, which is misleading. The operation on a per neuron level is completed only once the summation is also completed, which in their case doesn't happen at 120GHz since their scheme follows the TDM approach. They should instead use the term 120GOPS without the "per neuron" indication in the title, since this is their computational line-rate (line-rate of multiplications) but not the operational rate per neuron.

Response: Thank you for agreeing with that our work can be published. According to reviewer's suggestion, we have changed the term 120 GOPS/neuron in the manuscript to 120 GOPS.

Sincerely yours,

Lukas Chrostowski (on behalf of the co-authors)